# Automatic unsupervised respiratory analysis of infant respiratory inductance plethysmography signals

Carlos A. Robles-Rubio[1☯], Robert E. Kearney[1☯], Gianluca Bertolizio[2], Karen A. Brown[2☯]*

1 Department of Biomedical Engineering, McGill University, Montreal, Quebec, Canada, 2 Department of Anesthesia, Division of Pediatric Anesthesia, McGill University Health Centre, Montreal, Quebec, Canada

☯ These authors contributed equally to this work.
* karen.brown@mcgill.ca

**Data Availability Statement:** https://bitbucket.org/rkearney/aurea_distribution/src/master/.

**Funding:** The author(s) received no specific funding for this work.

## Abstract

Infants are at risk for potentially life-threatening postoperative apnea (POA). We developed an Automated Unsupervised Respiratory Event Analysis (AUREA) to classify breathing patterns obtained with dual belt respiratory inductance plethysmography and a reference using Expectation Maximization (EM). This work describes AUREA and evaluates its performance. AUREA computes six metrics and inputs them into a series of four binary k-means classifiers. Breathing patterns were characterized by normalized variance, nonperiodic power, instantaneous frequency and phase. Signals were classified sample by sample into one of 5 patterns: pause (PAU), movement (MVT), synchronous (SYB) and asynchronous (ASB) breathing, and unknown (UNK). MVT and UNK were combined as UNKNOWN. Twenty-one preprocessed records obtained from infants at risk for POA were analyzed. Performance was evaluated with a confusion matrix, overall accuracy, and pattern specific precision, recall, and F-score. Segments of identical patterns were evaluated for fragmentation and pattern matching with the EM reference. PAU exhibited very low normalized variance. MVT had high normalized nonperiodic power and low frequency. SYB and ASB had a median frequency of respectively, 0.76Hz and 0.71Hz, and a mode for phase of 4° and 100°. Overall accuracy was 0.80. AUREA confused patterns most often with UNKNOWN (25.5%). The pattern specific F-score was highest for SYB (0.88) and lowest for PAU (0.60). PAU had high precision (0.78) and low recall (0.49). Fragmentation was evident in pattern events <2s. In 75% of the EM pattern events >2s, 50% of the samples classified by AUREA had identical patterns. Frequency and phase for SYB and ASB were consistent with published values for synchronous and asynchronous breathing in infants. The low normalized variance in PAU, was consistent with published scoring rules for pediatric apnea. These findings support the use of AUREA to classify breathing patterns and warrant a future evaluation of clinically relevant respiratory events.

**Competing interests:** The authors have declared that no competing interests exist.

## Introduction

Apnea, the cessation of breathing, occurs in 80% of extreme premature infants, <30 weeks post menstrual age; decreasing to one per 1000 in healthy term infants [1, 2]. Anesthesia, when administered to young infants increases the incidence of apnea in the postoperative period; a clinical entity known as postoperative apnea (POA) [3, 4]. Both central and obstructive apnea are reported [5, 6]. Central apneas are characterized by an absence of inspiratory effort. Obstructive apneas exhibit respiratory effort but an absence of inspiratory airflow because of a blockage in the pharyngeal airway. Both central and obstructive apnea are life-threatening as medical intervention is required to re-establish breathing in the majority of infants; deaths have been reported [7–9].

The incidence of POA varies tenfold depending on the method used to detect apnea [3, 8]. These methods include bedside observation, critical event recorders, and the continuous acquisition of cardiorespiratory signals. Retrospective chart reviews detecting apnea from case notes, report POA in 10.5% of premature infants [10]. The GAS (General Anesthesia compared to Spinal anesthesia) Consortium, also detecting apnea from case notes, reported an incidence of POA ranging from 0.3 to 6% [9]. A prospective study of 91 premature infants, detecting apnea with a critical event recorder, reported an incidence of 10.1% [7]. Smaller prospective studies of infants, detecting apnea from continuous recordings of cardiorespiratory signals, reported POA in 40% to 62% of the cases [5, 11]. Thus continuous recordings of respiratory signals offer a sensitive method to detect apnea.

Apnea are often preceded by a disruption in breathing such as short respiratory pauses, sighs and hypoventilation [5, 12, 13]. Moreover, a postoperative breathing pattern with a high density of respiratory pauses was associated with POA [5]. These observations suggest that an analysis of breathing patterns might identify a biomarker for POA.

Dual-belt respiratory inductance plethysmography (RIP), recording the excursions of both ribcage and abdomen, is recommended by the American Academy of Sleep Medicine (AASM) to detect both breathing effort and thoracoabdominal synchrony [14]. The accepted method to analyze RIP is manual scoring, according to rules, which represent the consensus of experts [14, 15]. Manual scoring suffers from intra- and inter- scorer variability [16–18] and is time consuming, limiting research on POA to small studies of 20 to 60 infants [5, 6, 11, 13].

To mitigate the above we developed an automated unsupervised system to both classify and analyze infant breathing patterns in signals recorded with dual belt RIP [19, 20]. We refer to this system as AUREA, an acronym for Automated Unsupervised Respiratory Event Analysis. In an exploratory study of 24 infants, detecting the PAU pattern with a prototype algorithm, infants were grouped using an unsupervised, clustering analysis according to the maximum duration of pause events [19, 20]. The authors reported a 14.6s threshold separating infants with and without apnea; a finding that is noteworthy as respiratory pauses exceeding 15s are not considered normal in term infants, [2] and a central pause ≥15s is the threshold used by many to define POA [3–5]. However as the performance evaluation of this automated system had not yet been conducted we did not know if this finding was credible.

The aims of the current work were two-fold: first to publish a full description of AUREA and second to evaluate its performance. To this end we developed ground truth data with a procedure based on expectation maximization, which combined the patterns assigned in multiple manually scored records [21]. This procedure was published as the Expectation Maximization Pattern Sequence (EM-PSEQ) method; in the current work we abbreviate it to EM. As the usual method to analyze breathing patterns in studies of POA has been manual scoring by a single expert, [5, 6, 11] we also compared the classification performance of AUREA with that of an individual human scorer.

## Materials and methods

### Ethics statement

The dataset used to conduct this performance evaluation comprised 21 of the 24 datafiles previously reported by Robles-Rubio et al. [19–21]. Data were acquired from infants in the surgical recovery room at the Montreal Children's Hospital between March 2009 and April 2012. The study was approved by Institutional Review Board of the McGill University Health Center / Montreal Children's Hospital (Study 13-427-PED) and was conducted in accordance with Good Clinical Practice Guidelines and Standard Operating Procedures. Parents were informed of the study in advance of surgery. On the day of surgery written informed consent was obtained from a parent for each infant. The perioperative management of the infants was not standardized. Four signals were recorded: two RIP signals RCG and ABD, and in addition the photoplethysmography (PPG) and saturation signals obtained from oximetry. In addition, the Institutional Review Board of the McGill University Health Center / Montreal Children's Hospital approval (Study 12-308-PED) was obtained for the recruitment and training of 3 human scorers to manually classify the breathing patterns.

Signals were recorded continuously at a sampling rate of 50Hz/channel, over 5 to 12 hours of observation during which the infants exhibited a range of behavior including feeding, crying, and sleeping. The four signals were preprocessed to insert twice, at random locations, a balanced selection of reference segments representative of the breathing patterns [21–23]. The total length of the reference signals inserted into each record was 2,000s (about 10% of the total record length) and the reference segments were used in the development of the EM record as described by Robles-Rubio et al. [21]. Representative raw signals for RCG, ABD, PPG, and saturation are shown in the top four panels of Fig 1.

### Description of the automated unsupervised respiratory event analysis

**Overview.**  AUREA is an unsupervised classification system that assigns to each sample one of 5 unique patterns: pause (PAU), movement (MVT), synchronous breathing (SYB), asynchronous breathing (ASB), or unknown (UNK). This classification is performed in two stages as illustrated in Fig 2. In the first stage, a series of six metrics are computed from the RIP signals RCG and ABD to characterize their instantaneous amplitude, frequency content, and instantaneous phase relations. In the second stage, AUREA uses a series of binary k-means classifiers to assign each sample to a pattern that is determined by combining the output of the pattern detectors as a decision tree. The decision tree engages the pattern detectors, sequentially to detect the class/anti-class pairs. If the decision is set to the target class, the tree has reached a leaf and the classification is over. If the decision is set to the anti-class, the sample moves to the next tree node where another binary classification is performed. The order of the class/anti-class pairs in this decision tree is PAU, MVT, SYB, ASB. The PAU detector has the highest precedence and is engaged first; the other detectors are disengaged. The rationale for the high precedence for PAU detector was that during pauses the power in the RIP signals is very low and as a result the variance of the MVT, SYB, and ASB metrics would be high leading to many false positives. The MVT detector was engaged next since the low frequency power associated with movement would bias the SYB and ASB detectors. Each of these pattern detectors and stages will be discussed separately.

**AUREA metrics.**  *Analysis windows*. Metrics were computed using rectangular analysis windows centered around a time n. The analysis window advanced by one sample at a time to provide estimates of the metrics for each sample [19, 24, 25]. The width of the analysis window varied for each detector as reported in Table 1.

**Fig 1. Representative signals.** The raw signals of ribcage, abdomen, photoplethysmography (PPG), and saturation are shown. The data has been scored by twice by three scorers (IS) in independent instances. The breathing pattern classifications by expectation maximization (EM) and the automated unsupervised respiratory event analysis (AUREA) are also shown. Consecutive samples with identical breathing patterns are displayed as pattern segments. Both intra- and inter- scorer variability is evident. AUREA classified the SIH pattern as MVT; located in segments at 7s and 30s. Fragmentation is evident in the data classified by AUREA between 43s and 50s. Pattern legend ASB = asynchrony, MVT = movement, PAU = pause, SIH = sigh, SYB = synchrony, UNK = unknown.

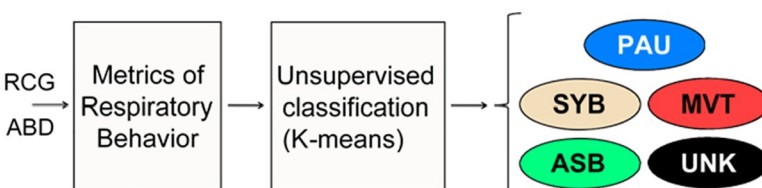

**Fig 2. Overview of AUREA.** Metrics of respiratory behavior are estimated from the ribcage (RCG) and abdomen (ABD) signals. These metrics are then input sequentially into a series of binary k-means classifiers to assign each sample to one of five patterns. respiratory pause (PAU), movement artifact (MVT), synchronous-breathing (SYB), asynchronous-breathing (ASB), unknown (UNK).

**Table 1. Analysis windows.** Analysis window widths used in each of the stages of signal processing for the Automated Unsupervised Respiratory Analysis (AUREA). The table gives the width of each analysis window width and provides a rationale for this choice.

| Window | Analysis Window Width (s) | Description of and Rationale for Analysis Window Width |
|---|---|---|
| | | **Preprocessing of RIP signals** |
| $N_{LF}$ | 5 | Analysis window width used for initial high-pass filtering of RIP signals. Based on previous work [19, 25]. |
| | | **PAU Metric** |
| $N_{QV}$ | 120 | Analysis window width for computing the quantiles. We observed that infants have a SYB pattern more than 60% of the time. Thus, we considered that a normalization quantile equal to the median should provide an appropriate reference to normal SYB values and a normalization analysis window of 2 minutes. |
| $N_V$ | 1 | Analysis window width for the estimate of variance; chosen empirically to provide for the fast response needed to detect short respiratory pauses. |
| $N_{DT}$ | 5 | Analysis window width for the removal of low frequency components. Based on previous work [19, 25]. |
| | | **MVT Metrics** |
| $N_{MA}$ | 1.42 | Analysis window width used for the moving average notch filter. This width was selected such that the filter nulls were at harmonics of the most frequent respiratory frequency, which we defined as the mode of the respiratory frequency histogram, computed from the entire data set. This frequency was 0.7 Hz or 42 breaths per minute. |
| $N_{QRMS}$ | 600 | Analysis window width for computing the quantiles variance in the RIP signals. Based on results in ref [19, 25]. |
| $N_{RMS}$ | 5 | Analysis window width for computing RMS values of RIP signals. Based on results in ref [19, 25]. |
| $N_{DT}$ | 5 | Based on results in ref [19, 25]. |
| | | **SYB and ASB Metrics** |
| $N_{SMO}$ | 0.42 | Analysis window width for the smoothing window used in computing SYB and ASB metrics. Based on results in ref [24]. |
| $N_B$ | 2 | Analysis window width used to compute the power of the SUM and DIFF signals. Based on results in ref [24]. |
| $N_{DT}$ | 2 | Based on results in ref [24]. |

ASB = asynchrony, PAU = pause, MVT = movement, SYB = synchronous.

*Preprocessing of signals*. The RIP signals were preprocessed to remove the mean value and low frequency components with a segment level mean normalization using an analysis window of width $N_{LF}$ [26].

$$RIP_{LF}(n) = \frac{1}{N_{LF}} \sum_{i=n-N_1}^{n+N_1} RIP_{RAW}(i)$$

where :

$RIP_{RAW}$ = raw RIP signal (RCG or ABD)

$N_{LF}$ = width of low frequency window (samples)

$$N_1 = \frac{N_{LF} - 1}{2}$$

(1)

and the preprocessed RIP signals were:

$$RIP(n) = RIP_{RAW}(n) - RIP_{LF}(n)$$

(2)

*Log normalized variance metrics*. The log normalized variance metric provides a measure of the instantaneous variance of a RIP signal normalized to its maximum. The AASM defines, a respiratory pause as a data segment where the RCG and ABD signals have amplitudes less than 10% of the preceding normal breath [14]. Consequently, it is expected that low values of the normalized variance of RCG and ABD would be discriminatory for the PAU pattern.

As the analysis window advanced one sample at a time, there was an estimate for the variance at every analysis window. The normalized variance metric of the RIP signal was computed from the estimated variance at each sample using an analysis window of width $N_V << N_{LF}$ as

$$V_{RIP} = \frac{1}{N_V} \sum_{n-N_{v2}}^{n+N_{v2}} (RIP(n))^2$$

where :

$\quad V_{RIP} =$ instaneous variance estimate

$\quad RIP =$ preprocessed RIP signal

$\quad N_V =$ width of variance window (samples)

$\quad N_{V2} = \frac{N_V - 1}{2}$

(3)

To compensate for nonstationarities in the amplitude of the RIP signals due to postural changes and/or slight displacement of the respibands, this value is then normalized as:

$$NV_{RIP}(n) = \ln\left(\frac{V_{RIP}(n)}{V_{RIP}^Q(n)}\right)$$

(4)

where $V_{RIP}^Q(n)$ is the $q^{th}$ quantile of the most recent $N_{qv}$ samples of $V_{RIP}(n)$.

*NonPeriodic power metric.* The MVT pattern is defined as segments where the RCG and ABD signals display a chaotic, non-sinusoidal, low frequency motion associated with active or passive movement of the infant [22]. Consequently we felt that a metric related to nonperiodic power (NPP), which we developed to detect movement artifacts in the PPG signal, [25] would be useful for detecting MVT. This metric is based on two assumptions: (i) in the absence of movement the RIP signal will have a quasi-periodic waveform; and (ii) during movement the RIP signal will contain stochastic nonperiodic noise whose amplitude is larger than that of the artifact-free signal [26, 27]. The NPP metric is computed as follows:

1. Compute the moving average of the RIP signal with an analysis window of width $N_{MA}$

$$RIP_{MA}(n) = \frac{1}{N_{MA}} \sum_{i=n-N_2}^{i=n+N_2} RIP(n)$$

where :

$N_2 = \frac{N_{MA} - 1}{2}$

(5)

This will produce a low-pass filtered version of the RIP signal with deep nulls at integer multiples of $\frac{f_s}{N_{MA}}$ where $f_s$ is the sampling frequency. If $N_{MA}$ is chosen such that these nulls occur at the respiratory frequency and its harmonics, the filter will attenuate periodic components related to respiration, pass other lower frequencies, and attenuate high frequency noise [25].

2. Compute the root mean square (RMS) value of $RIP_{MA}$ over an analysis window of width $N_{rms}$:

$$rms_{RIP_{MA}}(n) = \left(\frac{1}{N_{rms}} \sum_{n-N_2}^{n+N_2} (RIP_{MA}(n))^2\right)^{\frac{1}{2}}$$

where :

$N_{rms} =$ rms window width

$N_2 = \frac{N_{rms} - 1}{2}$

(6)

3. Normalize this by

$$npp_{RIP}(n) = \ln\left(\frac{rms_{RIP_{MA}}(n)}{rms^{Q}_{RIP_{MA}}(n)}\right) \qquad (7)$$

where $rms^{Q}_{RIP_{MA}}(n)$ is the $q^{th}$ quantile of $rms_{RIP_{MA}}(n)$ computed over the previous $N_{QRMS}$ samples where $N_{QRMS} >> N_{rms}$

*Synchronous breathing metric.* The synchronous breathing metric provides a measure of the synchrony between RCG and ABD

1. Smooth the preprocessed RIP signals to reduce additive noise with an analysis window of width $N_{SMO}$:

$$RIP_{SMO}(n) = \frac{1}{N_{SMO}}\sum_{i=n-N_2}^{i=n+N_2} RIP(i) \qquad (8)$$

where

$$N_{SMO} = \text{smoothing window width } (N_{SMO} << N_{DT})$$
$$N_2 = \left(\frac{N_{SMO} - 1}{2}\right)$$

2. Convert the smoothed signals to binary:

$$RIP_B = 1 \ if \ RIP_{SMO}(n) > RIP_{LF}(n)$$
$$= 0 \ otherwise \qquad (9)$$

where $RIP_{LF}$ is the low-frequency component.

3. Compute the sum and difference of the binary RCG and ABD signals as

$$SUM(n) = (RCG_{SMO} + ADB_{SMO})/2$$

and

$$DIF(n) = (RCG_{SMO} - ABD_{SMO})/2$$

When breathing is completely synchronous, *SUM* will oscillate between 0 and 1 at the respiratory frequency while *DIF* will stay constant at 0. Conversely, when breathing is asynchronous, *SUM* will remain constant at 0.5 while *DIF* will oscillate between -0.5 and 0.5 at the respiratory frequency [28].

4. High-pass filter *SUM* and *DIF* at 0.5 Hz to eliminate power associated with MVT and PAU [28]. This cut-off frequency was selected because most of PAU and MVT power is located between 0 Hz and 0.4 Hz [19, 26].

5. Estimate the power of the high-pass filtered sum signal, $SUM_{HP}$ with an analysis window of width $N_B$

$$b^+(n) = \frac{1}{N_B} \sum_{i=n-N_1}^{i=n+N_1} SUM_{HP}^2(i)$$

where

$$N_1 = (N_B - 1)/2$$

(10)

The resulting SYB metric, $b^+$, will have large values for SYB and tend to zero otherwise [28].

*Asynchronous breathing metric.* The asynchronous breathing metric is computed similarly except that it is based on the difference of the two binary signals. The ASB metric $b^-$ is the power of the high-pass filtered difference signal $DIF_{HP}$; its values will be high during ASB and tend to zero otherwise [28].

$$b^-(n) = \frac{1}{N_B} \sum_{i=n-N_1}^{i=n+N_1} DIF_{HP}^2(i)$$

where

$$N_1 = (N_B - 1)/2$$

(11)

*Respiratory frequency.* The instantaneous respiratory frequency $f_{RESP}$ is estimated using an approximate frequency demodulation technique based on zero crossings.

*Phase.* The phase ($\Phi$) between RCG and ABD RIP signals is estimated using the algorithm described by Motto et al. [29]. This involves two steps:

1. Compute the logical XOR of the binary signals:

$$\Phi_B = xor(RCG_B, ABD_B)$$

(12)

2. Compute the moving average of the result estimate the fraction of asynchrony as

$$\Phi[n] = \frac{1}{N_B} \sum_{i=n-\frac{N_B-1}{2}}^{n+\frac{N_B-1}{2}} \Phi_B[i]$$

(13)

Where $N_B$ = 101 samples. The $\Phi[n]$ metric is an estimate of the phase shift between the RCG and ABD signals. It assumes a value in the range [0 1] corresponding to [0 180] degrees.

**Training the k-means classifier.** The AUREA classifier is trained using the following steps shown schematically in Fig 3.

A. Use the entire training data set, and the RCG and ABD normalized variance metrics and pause metrics as features, to train a k-means classifier to distinguish two classes–one with low variance and one with high variance. Assign the low variance samples to the PAU pattern and remove them from the data set to generate a reduced data set.

B. Use the reduced training data set from step A), and the RCG and ABD nonperioidic power metrics as features, to train a k-means classifier to distinguish two classes–one with low nonperiodic power variance and one with high variance. Assign the samples with large

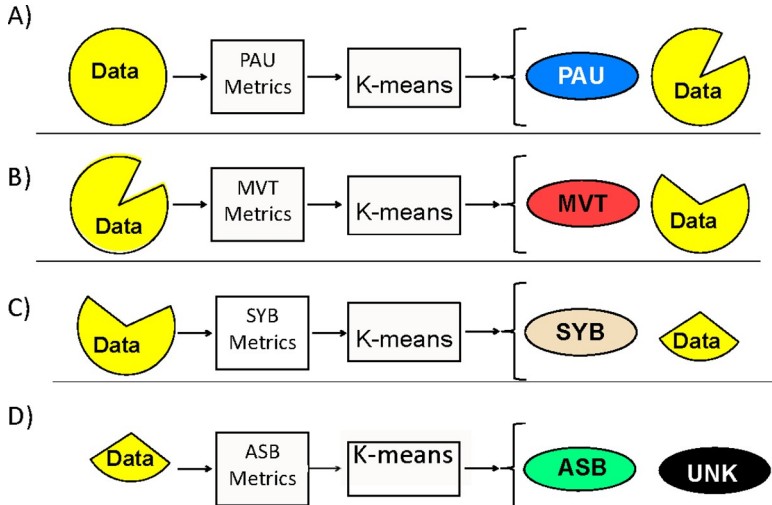

**Fig 3. Automated Unsupervised Respiratory Event Analysis (AUREA) training.** (A) The pause (PAU) metrics from the training data are input to a k-means classifier to classify PAU samples. (B) The movement artifact (MVT) metrics from the remaining data are input to k-means to classify MVT samples. (C) Then, the synchronous-breathing (SYB) metric is used with k-means to classify SYB samples in the remaining data. (D) Finally, the asynchronous-breathing (ASB) metrics are input to a k-means to discriminate between ASB and unknown (UNK) samples. The classification parameters of the 4 k-means classifiers are stored to use with new data.

nonperiodic variance to the MVT class and remove them from the data set to generate a reduced data set.

C. Use the reduced data set from Step B), and the synchronous breathing metric, to train a k-means classifier to distinguish two classes, with high and low values of the metric. Assign samples with large values to the SYB pattern, and remove them from the data set to generate a reduced data set.

D. Use the reduced data set from Step C), and the asynchronous breathing metric, to train a k-means classifier to distinguish two classes, with high and low values of the metric. Assign samples with large values to the ASB pattern, and remove them from the data set. Assign samples with low values of the asynchronous movement metric to the UNK pattern.

E. Store the classification parameters for the four classifiers for use with new data.

**Classification by AUREA.**   The procedure to classify data for a test subject proceeds as follows:

1. Compute the metrics from the RIP data.

2. Supply the metrics and parameters obtained from the training procedure to AUREA.

3. Classify data in the same sequence that was used during training using the k-means parameters estimated during the training phase to perform the classification. Consequently, once training is completed, classification is deterministic and very fast.

4. AUREA returns a Pattern SEQuence (PSEQ); a continuous signal that consists of a categorical sequence describing the pattern assigned to each sample of the original record. Fig 1 shows a representative segment of cardiorespiraory signals and PSEQ resulting from the AUREA analysis.

**Boundary adjustment.** Infant respiratory patterns are heavily unbalanced; PAU samples occur infrequently, representing $< 5\%$ of sample, while SYB samples are common, representing more than 60% of the samples [22]. In pilot studies, we found that k-means misclassified a significant number of the samples belonging to the most prevalent category. Consequently, we adjusted the k-means decision boundaries to mitigate the effects of the unbalanced data. To do so, we noted that in a dataset with $P$ input metrics, the k-means decision boundary between two clusters, $C_j$ and $C_m$, forms a hyperplane containing the point $\gamma_{jm} \in \mathbb{R}^P$ with the normal vector $\mathbf{v}_{jm} \in \mathbb{R}^P$ where

$$
\begin{aligned}
&\mathbf{v}_{jm} = \mathbf{c_m} - \mathbf{c}_j \\
&\gamma_m = w_{jm}\mathbf{v}_{jm} + \mathbf{c}_j \\
&\text{where} \\
&\mathbf{c}_j = \text{centroid of cluster} C_j \\
&\mathbf{c}_m = \text{centroid of cluster} C_m \\
&w_{jm} = \text{decision boundary weighting factor}
\end{aligned}
\tag{14}
$$

The decision boundary weighting factor, $w_{jm}$, determines the proportion of the Euclidean space covered by each cluster and for a conventional k-means implementation using a balanced dataset it has a value of 0.5. The assignment of sample $\mathbf{x}(n)$ to cluster $C_j$ termed $L\{\mathbf{x}(n)\}$, is determined as:

$$
L\{\mathbf{x}(n)\} = C_j \leftrightarrow v_{jm} \bullet (x(n) - \gamma_{jm}) < 0 \forall m \neq j
\tag{15}
$$

To adjust for the sample unbalance, we modified decision boundary weighting factor to reflect the relative sizes of the clusters once the k-means had converged:

$$
\begin{aligned}
&w'_{jm} = \frac{N_j}{N_j + N_m} \\
&\text{where} \\
&N_j = \text{number of samples in } C_j \\
&N_m = \text{number of samples in } C_m
\end{aligned}
\tag{16}
$$

This re-weighting shifts the decision boundary, as illustrated in reference [19], so that the cluster with more samples covers more space, thus mitigating the effect of sample unbalance.

**Implementation of AUREA.** AUREA is implemented as an object-oriented application in Matlab which is available at the web https://bitbucket.org/rkearney/aurea_distribution/src/master/. The metrics were computed using custom code, while k-means was carried out using the Matlab function *kmeans*.

## Ground truth data

AUREA is an unsupervised system that does not require ground truth data to operate. However, a ground truth data set is required to evaluate its performance. To generate ground truth data we recruited and trained scorers to manually score each record twice [22], and then used the EM procedure reported by Robles-Rubio et al. [21] to generate a consensus score. Each of these steps are now briefly described below.

**Manual scoring of raw data.** We used the RIP data set acquired from infants and RIP Score, a computer aided tool for manual scoring, described by Robles-Rubio et al. [22]. Using this tool, each scorer was first trained to manually score a data record and assign to each

sample one of six unique breathing patterns: pause (PAU), sigh (SIH), synchronous breathing (SYB), asynchronous breathing (ASB), movement (MVT) or unknown (UNK). The training procedure incorporated methods for assessing scorer accuracy and consistency, and training continued until a scorer attained an adequate level of performance. Once this was achieved, each scorer manually analyzed each preprocessed data record twice. Records were presented in random order for each analysis to minimize the possibility of bias due to prior exposure. The manual scoring of each record produced a continuous PSEQ containing the pattern assigned to each sample. The plots labeled IS(1)->IS(6) in Fig 1 illustrate the PSEQs obtained from this manual scoring.

The raw and manually scored data are available publicly from Dryad Digital Repository [22] and the RIPscore source code from software, from https://doi.org/10.1371/journal.pone.0134182.s010.

**Expectation maximization reference.** As illustrated in Fig 1, the 6 manual PSEQs exhibited intra- and inter-subject variability. Consequently, we combined them to generate a consensus PSEQ using an EM algorithm [21]. This method combines multiple, manual analyses to estimate the maximum likelihood breathing pattern. This is achieved by weighting the contributions from each manual analysis according to its estimated performance. EM is used to iteratively refine the pattern estimates and scorer performance. Simulation results indicated that with three scorers analyzing each record twice, the EM record should have a median kappa value of 95% [21]. Consequently, we considered the EM record as ground truth data with which to evaluate the classification performance of AUREA. The trace labeled EM in Fig 1 shows a representative PSEQ derived from the six IS PSEQs.

## Evaluation methods

**Merging of categories.** AUREA assigned five unique breathing patterns whereas EM assigned six; as AUREA did not attempt to classify the SIH pattern. Analysis of the confusion matrix showed that AUREA classified the 83.6% of SIH samples as MVT (77.0%) or UNK (6.6%). Since, there seemed little clinical reason to differentiate between MVT and UNK we merged the SIH, MVT, and UNK patterns into a single pattern, UNKNOWN.

**Performance metrics.** The performance of AUREA was evaluated by comparing its classifications to that of EM, sample by sample. To this end we generated a 4x4 confusion matrix, $M(i,j)$, in which each element corresponded to the number of samples of the EM pattern $i$ that AUREA assigned to pattern $j$. Thus, each column indicates how the samples assigned by AUREA to a specific pattern were assigned by EM. Conversely, the rows indicate the distribution of AUREA classified patterns for each EM pattern. Consequently, the diagonal elements of the confusion matrix correspond to correct assignments while the off-diagonal elements are incorrect assignments. Interpretation of a confusion matrix can be difficult. Therefore, in addition, we determined the true positives (TP) and negatives (TN) as well as false positives (FP) and negatives (FN) and calculated 4 performance criteria: overall accuracy, and pattern specific precision, recall, and F-score.

*Accuracy*. Overall accuracy was defined as the percentage of samples correctly identified and was defined as follows:

$$A = 100 * \frac{\sum_{i=1}^{4} M(i,i)}{\sum_{j=1}^{4}\sum_{i=1}^{4} M(i,j)} \tag{17}$$

Pattern specific performance was evaluated with precision, recall, and F-score.

*Precision*. The pattern specific precision provides a measure of the fraction of samples assigned to a pattern that were correct. That is:

$$P(i) = \frac{M(i,i)}{\sum_{j=1}^{4} M(i,j)} \tag{18}$$

*Recall*. The pattern specific recall provides a measure of the fraction of the samples of pattern $i$ that were correctly assigned to that pattern. That is:

$$R(i) = \frac{M(i,i)}{\sum_{j=1}^{4} M(j,i)} \tag{19}$$

*F-score*. The F-score is the harmonic mean of precision (P) and recall (R) and provides a pattern specific measure of performance. Thus:

$$F_1(i) = 2\frac{P(i)R(i)}{P(i) + R(i)} \tag{20}$$

**Statistical analysis.** Values were reported as mean and standard deviation (SD), median or mode as indicated. For all performance indices, higher values are better.

## Results and discussion

Data from 21 of the 24 infants reported by Robles Rubio et al. were used for this work; as three files were of unsatisfactory quality. Thus, records from 21 infants (male = 16, birth age of 31 ± 4 weeks) were analyzed. At the time of surgery, the postmenstrual age was 43 ± 2 weeks; the weight was 3.6 ± 1.0 kg. Nineteen of the infants were formerly premature infants (birth age < 36 weeks).

### Classification time

We used a Windows 10 computer with an Intel Core i9-9900K CPY @ 3.60GHz and 65bytes of memory to evaluate the AUREA's computation load. Training and classification the 21 records, corresponding to 116 hours of recording, took less than 35s and classification no more than 4s. This contrasts with an estimated 116 hours required for manual analysis; based on Robles Rubio et al. [22] report that the rate of scoring by human experts was one hour per hour of recorded data.

### Properties of patterns classified by AUREA

We were interested in how the properties of the samples AUREA assigned to each pattern compared to what would be expected based published scoring rules.

Fig 4 shows that the PAU samples have a normalized variance approximately two standard deviations lower than that of all other samples. This very low variance is consistent with the absence of inspiratory effort and airflow specified in the rule for pediatric apnea published by the AASM [14]. A similar probability distribution was observed for the RCG normalized variance.

Next we compared the properties of samples assigned to the MVT pattern to those of all other samples excluding PAU. Fig 5A shows the normalized nonperiodic power for MVT pattern was about 2 standard deviations higher than that of the other samples. Fig 5B shows the instantaneous frequency was about 0.5Hz lower. These features are consistent with the high

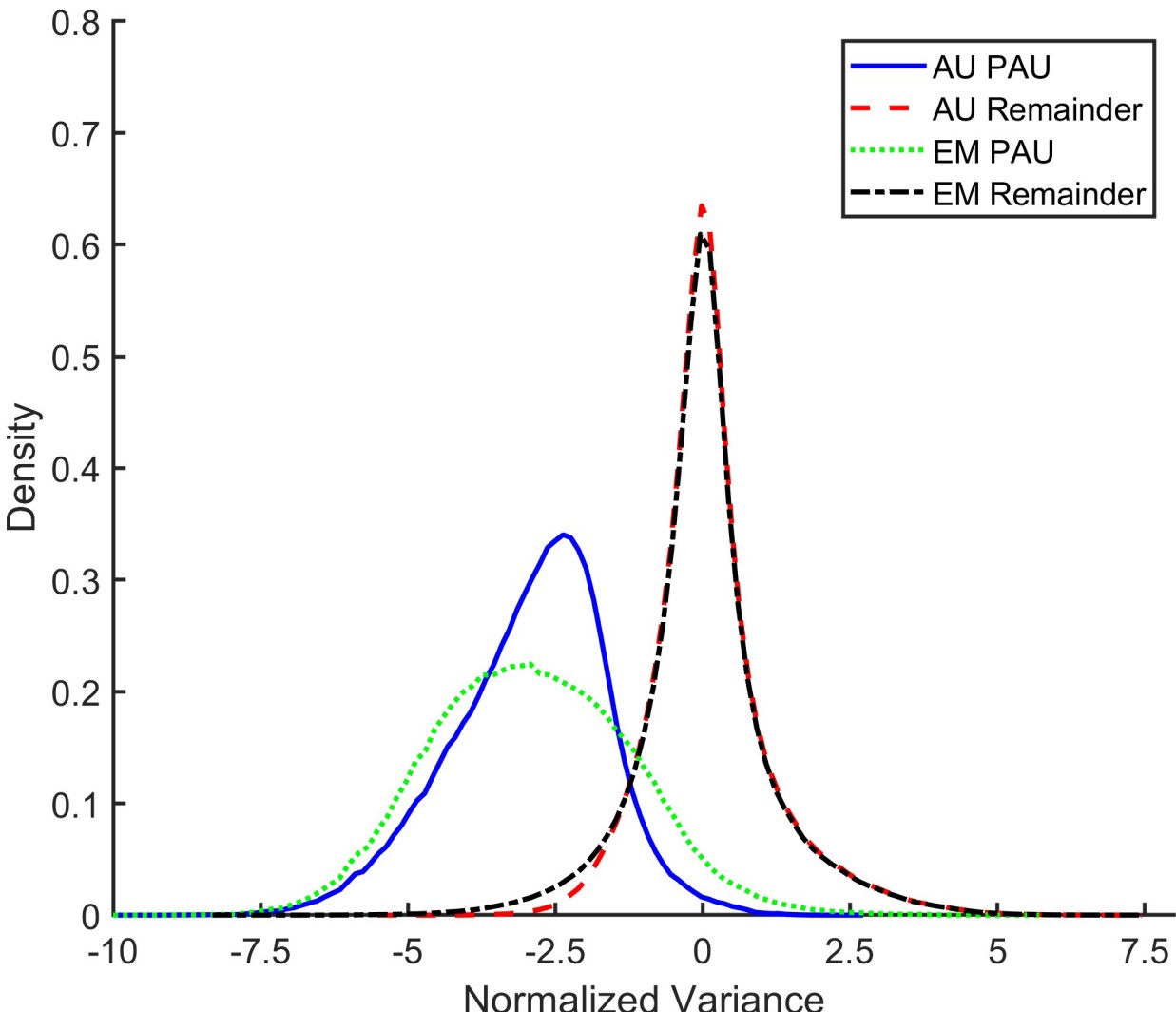

**Fig 4. Features of PAU samples.** Probability density of the normalized variance in the abdomen signal for samples classified as pause (PAU) compared with the remaining samples. PAU samples exhibited a very low normalized variance. AU = AUREA (Automated Unsupervised Respiratory Event Analysis), EM = Expectation Maximization.

amplitude, chaotic excursions observed in movement artifact. The RCG normalized variance and RCG frequency behaved similarly.

Fig 6 shows the distribution of instantaneous respiratory frequency for SYB, median of 0.76Hz [46 breaths per minute (bpm)] and ASB median 0.71 Hz (43bpm). These frequencies were consistent with published values in term infants during sleep, where respiratory rates range from 37 to 75 bpm [30, 31].

Fig 7A shows that the probability density of the instantaneous phase for SYB was skewed to the left (mode 4$^\circ$). In contrast Fig 7B shows that the curve for ASB was skewed to the right (mode 100$^\circ$).

Fig 8 illustrates the features of samples assigned to the UNK pattern. The ABD signal exhibited a median variance ~0, median nonperiodic power ~1, and a median instantaneous frequency ~0.5Hz; values overlapping those observed for PAU and MVT. Values for instantaneous phase overlapped the probability densities of SYB and ASB. A similar pattern was seen in the RCG.

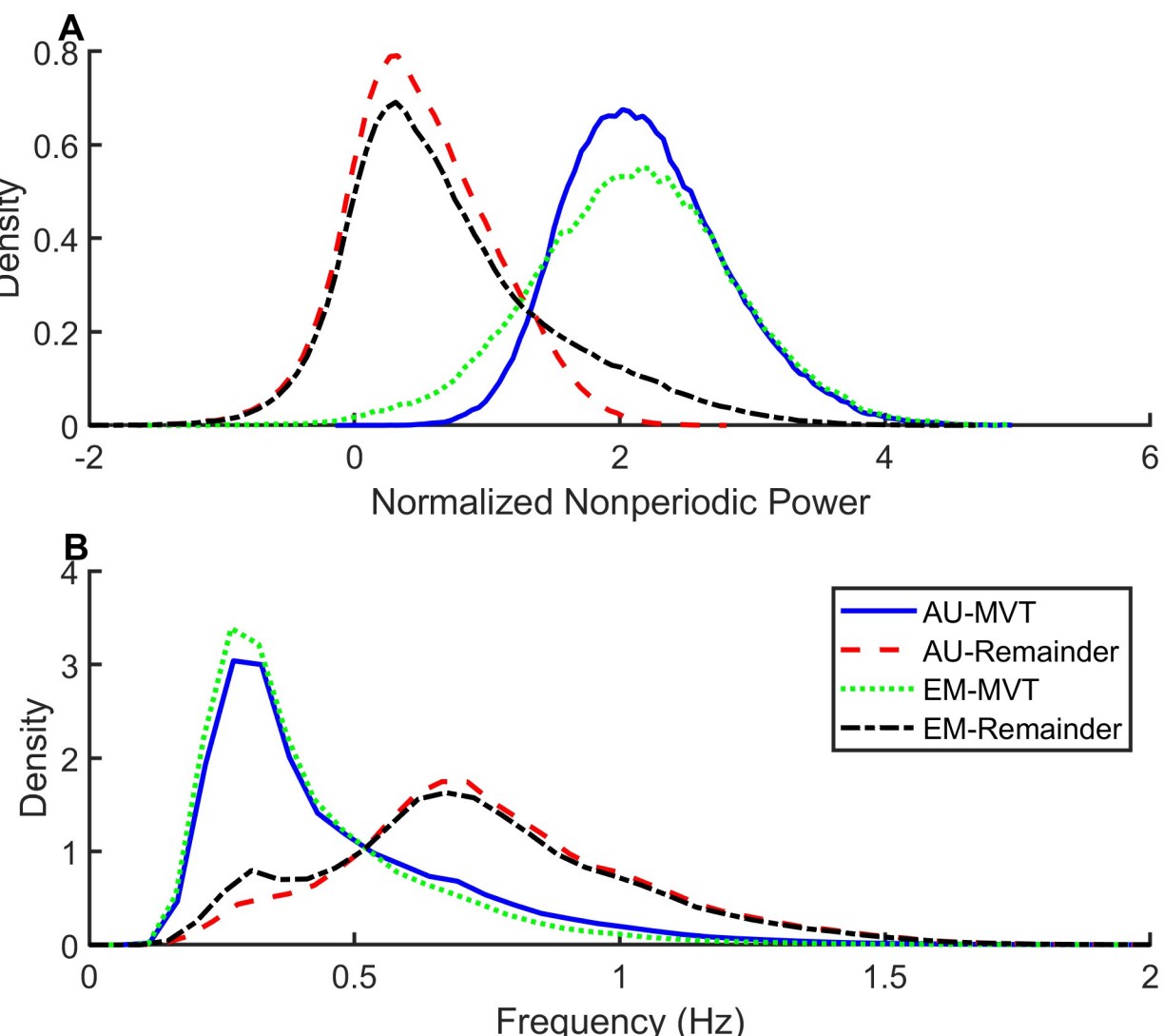

**Fig 5. Features of MVT samples.** The probability density of the (A) Normalized nonperiodic power and (B) instantaneous frequency n the abdomen signal for samples classified as movement (MVT) compared with the remaining samples. MVT samples exhibited high normalized nonperiodic power and low instantaneous frequency. The thresholds for normalized nonperiodic power separating MVT from the remaining samples were lower for EM compared with AUREA. AU = AUREA (Automated Unsupervised Respiratory Event Analysis), EM = Expectation Maximization.

The distribution of patterns classified by AUREA was unbalanced and SYB was the most frequent pattern. (Fig 9)

## Properties of patterns classified by expectation maximization

Features of the PAU samples classified by EM were similar to those AUREA with very low normalized variance; approximately two standard deviations below the remaining samples. (Fig 4) A similar probability distribution was observed for the RCG normalized variance. The threshold to classify PAU was wider for samples classified by EM compared with AUREA. The very low normalized variance in the PAU pattern is consistent with the rule reported by Weese-Mayer et al. to classify apnea in the CHIME (Collaborative Home Infant Monitoring Evaluation) data, namely an amplitude of the RIP sum waveform <25% of the baseline [15]. It is also

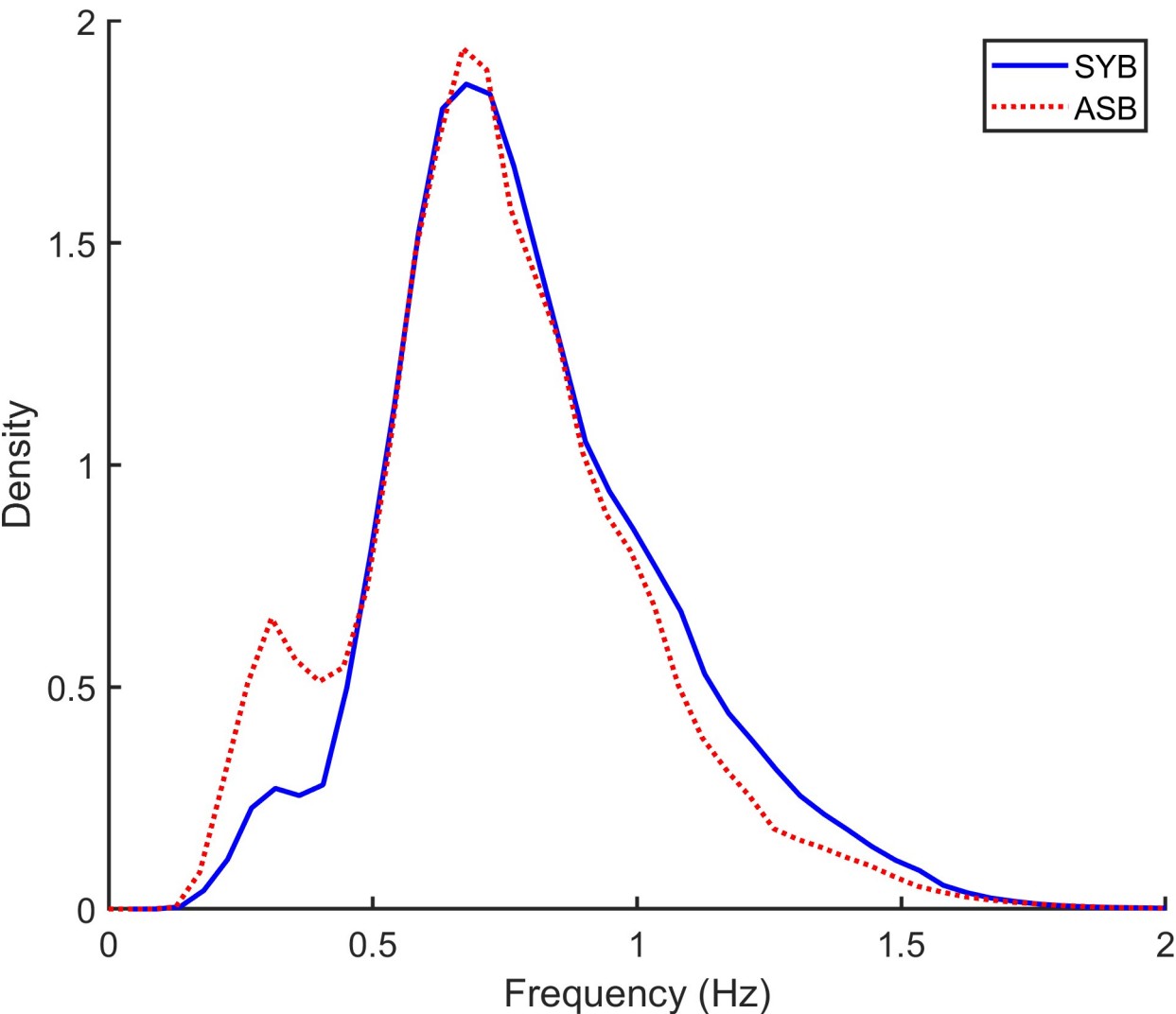

**Fig 6. Breathing frequencies.** The probability density for the instantaneous frequency for synchronous (SYB) and asynchronous (ASB) samples classified by the automated respiratory event analysis (AUREA).

consistent with AASM scoring rule 5.2 defining a pediatric apnea as "a drop in the peak signal excursion by ≥90% of the pre-event baseline'" [14].

In EM classified MVT samples, the normalized nonperiodic power was 2 standard deviations higher and the instantaneous frequency was about 0.5Hz lower than the remaining samples. (Fig 5) A similar probability distribution was observed for the RCG. However the threshold values were wider for EM classified MVT patterns compared with AUREA. The AASM provides scoring rules for limb movement, but is silent on scoring rules for movement artifact corrupting the RIP signals [14].

In a manual analysis of the CHIME (Collaborative Home Infant Monitoring Evaluation), Weese-Mayer et al. defined an obstructed breath as out-of-phase rib cage and abdomen RIP signals, but did not define the phase threshold for this classification [15]. Similarly, the AASM scoring rules are silent on the phase threshold(s) defining synchronous, asynchronous and paradoxical breathing patterns. Thus we interpreted values for the instantaneous phase according to the intervals suggested by Allen et al.: synchronous breathing = zero degrees,

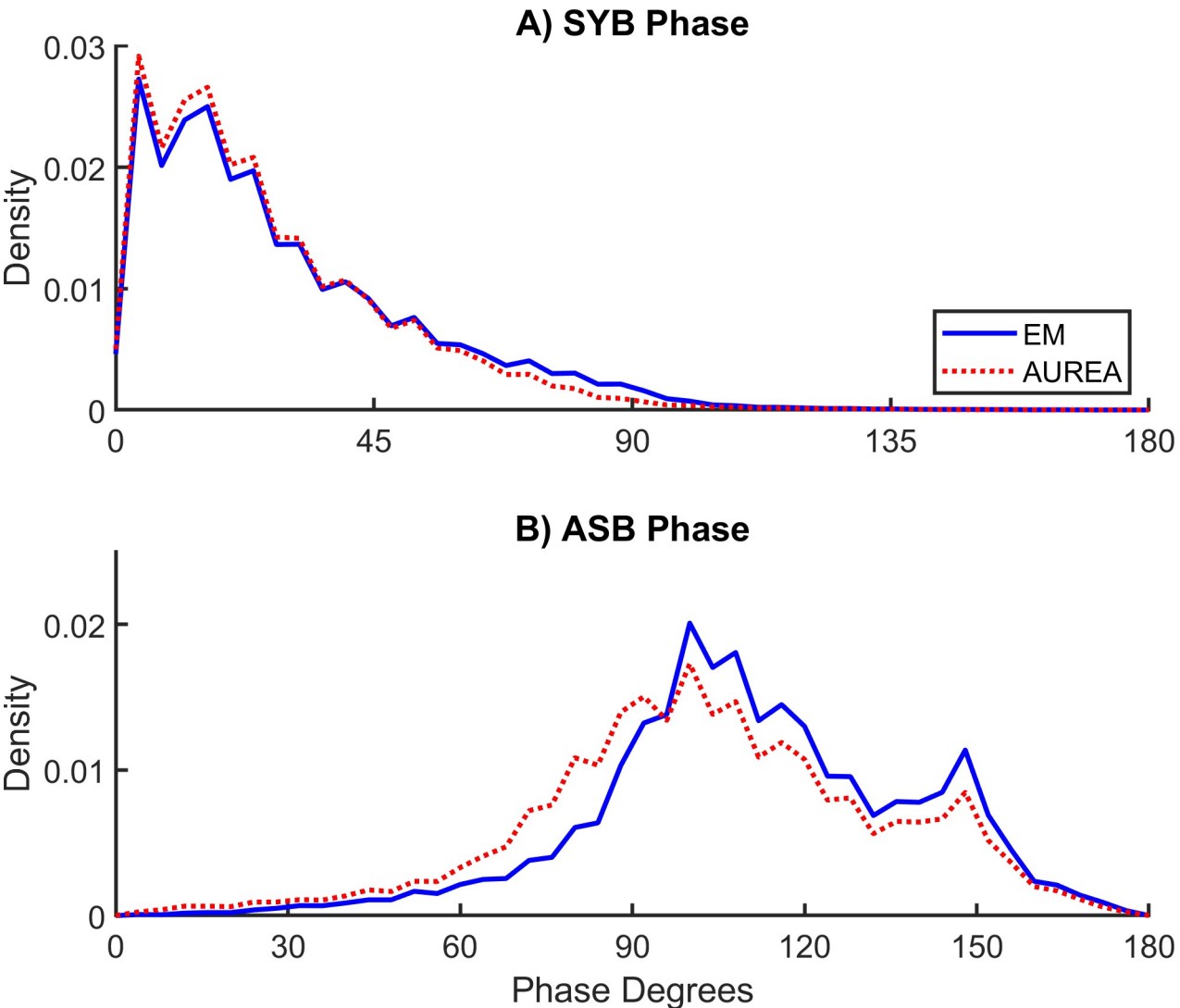

**Fig 7. Instantaneous phase for synchronous (SYB) and asynchronous (ASB) samples.** The threshold separating ASB from SYB samples was about 10° lower for the unsupervised respiratory event analysis (AUREA) compared with classification by expectation maximization (EM).

thoracoabdominal asynchrony 45 < 135 degrees, and paradoxical breathing = 180 degrees [32]. In the manual scoring of the RIP signals, evaluation of thoracoabdominal asynchrony was aided by the Lissajous loop (X-Y plot) displayed by the software RIPScore [22]. A slope ≤ 0 was the criteria for the breathing pattern ASB. The threshold for classification by AUREA was determined empirically by the k-means classifier. Fig 7 shows that threshold discriminating ASB and SYB was about 10 degrees lower for AUREA (64°) compared with EM (73°).

In a study of POA, Brown et al., reported the threshold separating synchronous and asynchronous breathing patterns was 54 degrees [5]. For both the AUREA and EM classifications, values for SYB instantaneous phase were consistent with phase angles reported from the manual analysis of Lissajous figures during quiet sleep in young infants. In 6 full-term neonates Allen et al. reported a phase angle between zero and fifteen degrees [32]. Warren et al., evaluating 7 term newborns, reported a phase angle between 6 and 18 degrees [31]. Degras et al., evaluating 8 term infants, reported the mean phase angle of 9.3 degrees [33].

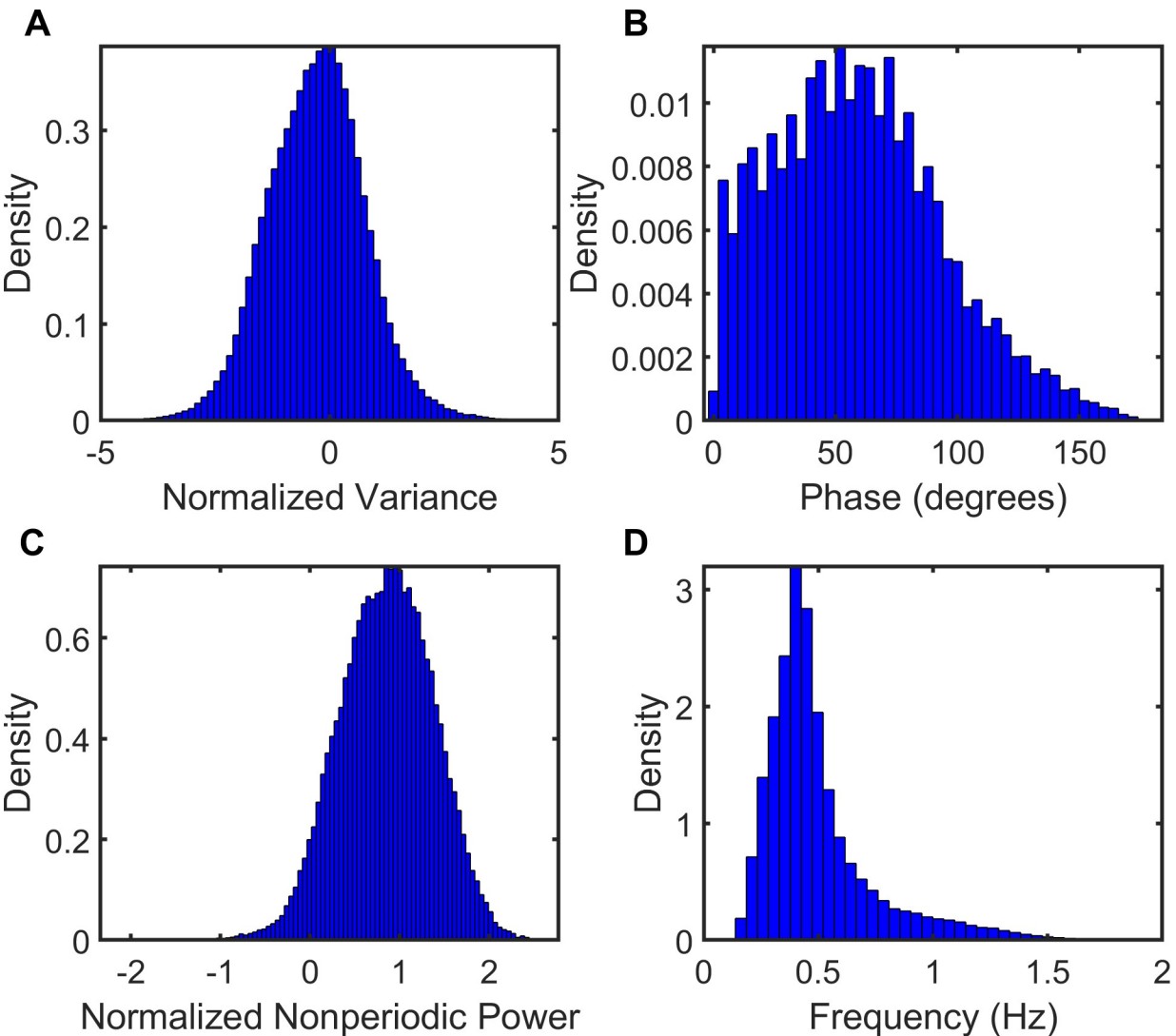

**Fig 8. Unknown (UNK) pattern.** Features of UNK in samples obtained from the abdomen signal which were classified by automated unsupervised respiratory event analysis (AUREA). ABD = abdomen, NPPNORM = normalized nonperiodic power.

The estimate of phase by AUREA has several advantages over the manual analysis of Lissajous loops. The manual analysis requires selection of ideal breaths with quasi-sinusoidal waveforms, specifically avoiding breaths with changes in behavioral state. In the studies cited above, this limited the estimate of phase angle from manual analysis to an evaluation of 10 to 40 breaths. In contrast, the phase estimate by AUREA requires neither an estimate of the breathing frequency nor breath segmentation. It is robust in the presence of signal drift. Moreover, it makes no assumption of a sinusoidal waveform, a constant breath frequency, or a constant breath amplitude.

## Performance evaluation of AUREA with expectation maximization

**Analysis of pattern samples.** In total 22,515,980 samples were classified. AUREA classified more samples as PAU and ASB than EM. (Fig 10A and Table 2)

The E-step in the EM procedure calculates the probability that the EM pattern is correct, thereby providing a measurement of the confidence in the EM estimate. In 90.6% of samples

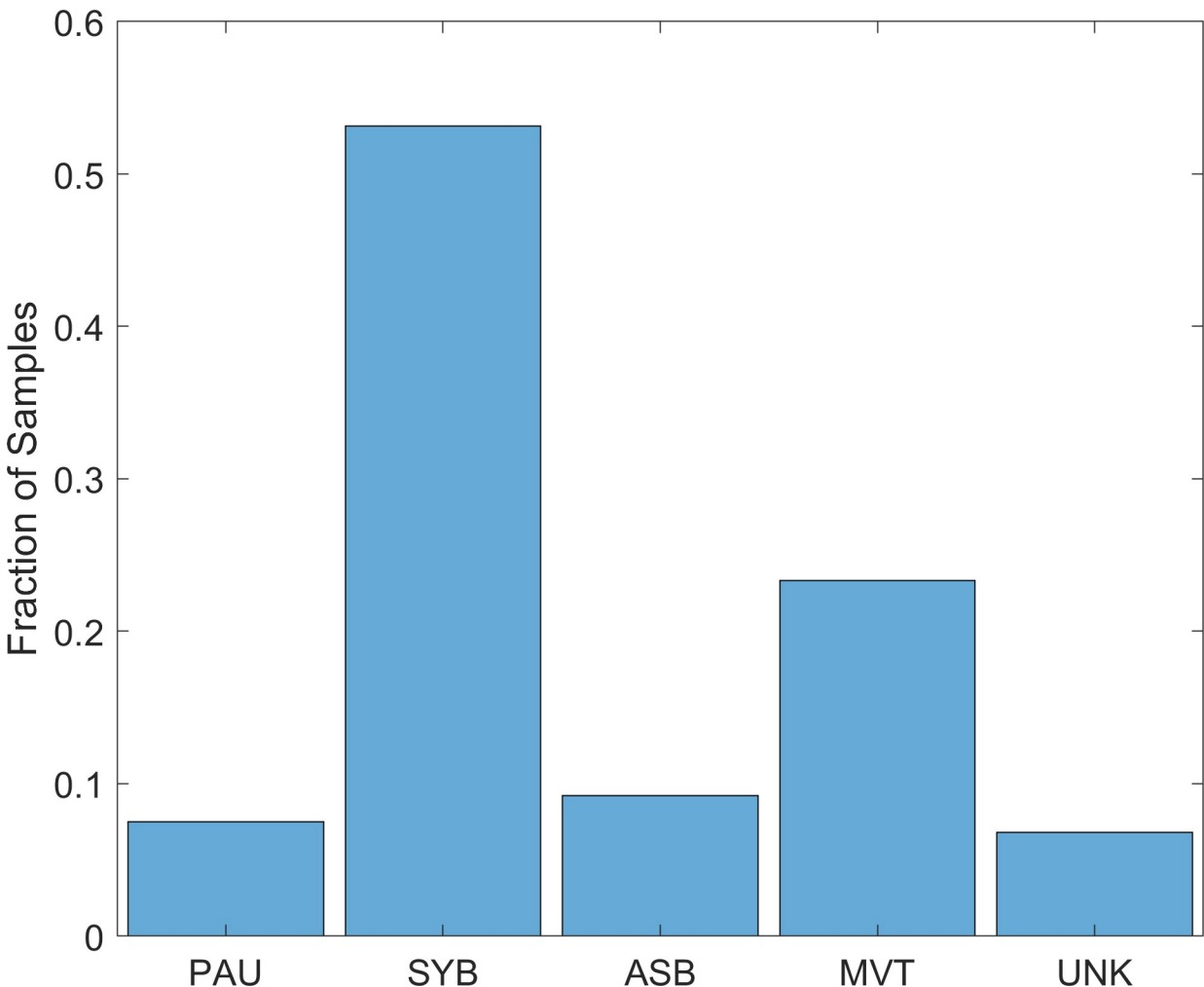

**Fig 9. Distribution of patterns.** Fraction of samples classified by the automated unsupervised respiratory event analysis (AUREA) for the 5 unique breathing patterns. ASB = asynchrony, MVT = movement, PAU = pause, SYB = synchronous, UNK = unknown.

the probability that the breathing pattern was correct was $\geq 0.95$. Confidence was highest for the pattern SYB and lowest with UNKNOWN. (Table 3) Only 75.68% of UNKNOWN samples had a >95% probability that the UNKNOWN pattern was correct, indicating a lower confidence in the EM estimate of the UNKOWN pattern.

The confusion matrix (Table 4) shows the pattern specific confusion was lowest for the SYB pattern (16.3%). The PAU pattern was confused most often with UNKNOWN (15.2%), followed by SYB (5.7%). The highest confusion was observed with UNKNOWN (25.0%), which was confused with all other patterns SYB>PAU>ASB, respectively 9.99%, 8.24%, and 7.30%.

Sample by sample, the mean overall accuracy was 0.80, ranging, across cases, from 0.66 to 0.87. (Fig 11) An accuracy less than 0.8 was observed in 8 of the 21 cases (Case 4,5,6, 13,14, 18, 20 and 21).

Across the 21 cases, precision was higher than recall for PAU and ASB. Recall was higher than precision for SYB. Mean performance indices are reported in Fig 12 and Table 5.

Across the 21 cases, inhomogeneities in pattern proportion were evident. Case 21 had many more PAU samples than the other cases. Cases 6, 14, and 21 had many more ASB

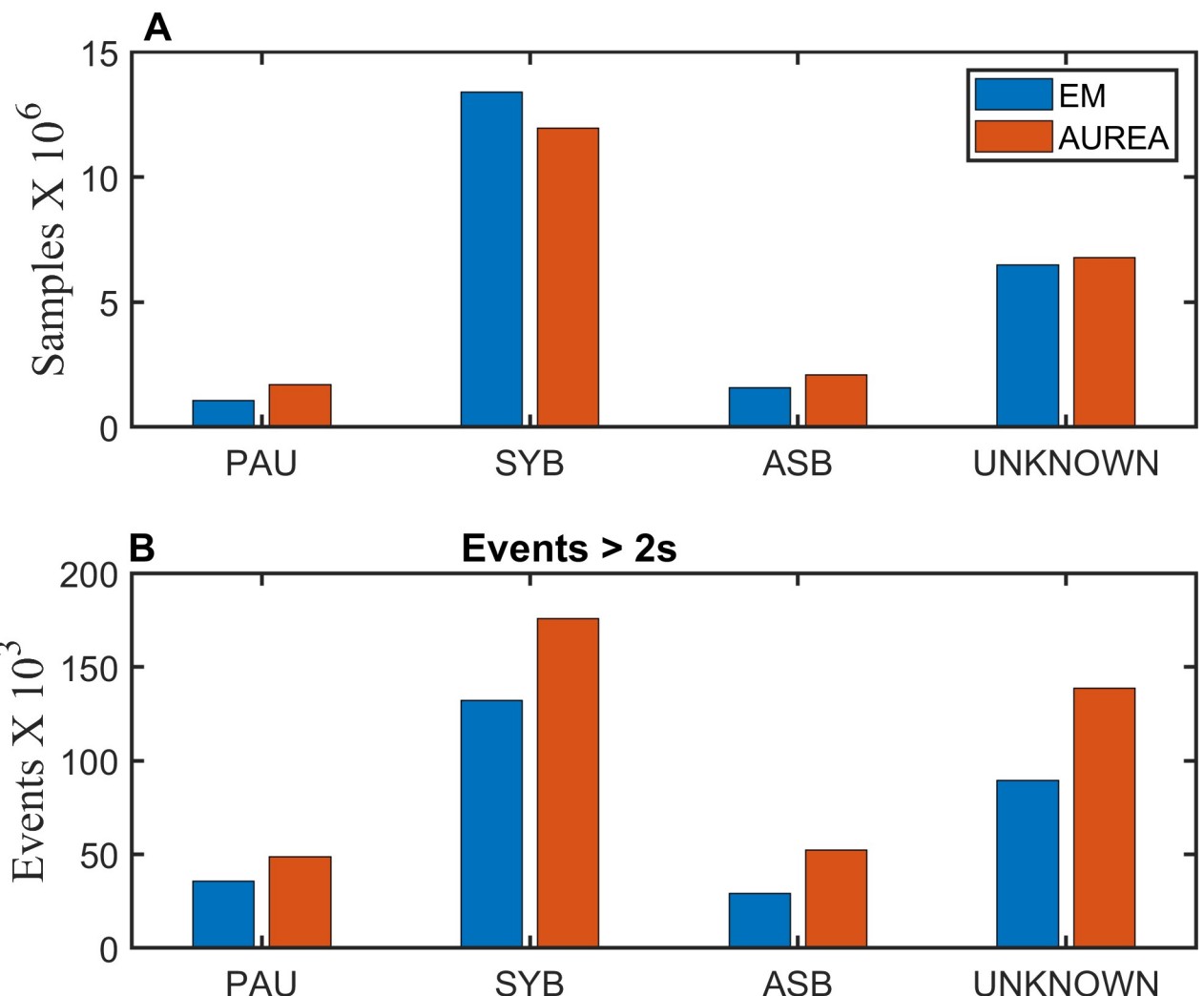

**Fig 10. Distribution of patterns.** *Fraction of samples classified into 4* unique breathing patterns by expectation maximization (EM) and AUREA (Automated Unsupervised Respiratory Event Analysis). A shows the sample by sample classification. B shows the classification for event lengths >2s. ASB = asynchrony, MVT = movement, PAU = pause, SYB = synchronous.

samples, and fewer SYB patterns. Classification by AUREA mirrored these inhomogeneities. (Fig 13)

**Table 2. Pattern classification.** Distribution of patterns for samples that were classified by Expectation Maximization (EM) and the Automated Unsupervised Respiratory Event Analysis (AUREA). Pattern specific distribution [n (%total)] are shown. MVT, UNK and SIH were combined into a single group UNKNOWN.

| | Patterns | | | | | |
|---|---|---|---|---|---|---|
| | **PAU** | **SYB** | **ASB** | **MVT** | **UNK** | **SIH** |
| EM | 1,064,433 (4.7%) | 13,403,481 (59.5%) | 1,569,436 (7.0%) | 3,400,497 (15.1%) | 2,694,942 (12.0%) | 383,191 (1.7%) |
| AUREA | 1,688,600 (7.5%) | 11,966,554 (53.2%) | 2,078,901 (9.2%) | 5,251,091 (23.3%) | 1,530,834 (6.8%) | not classified |
| | PAU | SYB | ASB | UNKNOWN | | |
| EM | 1,064,433 (4.7%) | 13,403,481 (59.5%) | 1,569,436 (7.0%) | 6,478,630 (28.8%) | | |
| AUREA | 1,688,600 (7.5%) | 11,966,554 (53.2%) | 2,078,901 (9.2%) | 6,781,925 (30.1%) | | |

ASB–asynchrony, MVT–movement, PAU–pause, SIH = sigh, SYB = synchronous, UNK = unknown.

**Table 3. Confidence in expectation maximization estimate.** Proportion of samples with a ≥95% probability that the Expectation Maximization (EM) estimate was correct.

| PATTERN | Proportion of Samples with >95% Probability that the EM estimate was correct (%) | Mean Probability | Standard Deviation of Probability |
|---|---|---|---|
| Overall | 90.64 | 0.97 | 0.07 |
| PAU | 90.70 | 0.97 | 0.08 |
| SYB | 97.70 | 0.99 | 0.03 |
| ASB | 91.99 | 0.98 | 0.07 |
| UNKNOWN | 75.68 | 0.94 | 0.10 |

**Analysis of pattern events.** At event lengths below 2 seconds (100 samples), both AUREA and EM exhibited fragmentation. Fig 14 shows that compared with EM AUREA detected many more very short event segments with fewer than 50 samples.

*Pattern matching.* At segment lengths ≥100 samples (2s), 75% of the EM determined pattern events had >50% of the samples assigned the same pattern by AUREA. In addition, 65% of the EM determined pattern events had >75% of the samples assigned the same pattern by AUREA. Pattern specific matching was highest for the PAU pattern. (Table 6) Moreover, 22 of the 24 (92%) PAU events ≥14s classified by AUREA had more than 50% of samples also assigned the PAU pattern by EM.

**Performance evaluation of AUREA with an individual scorer.** Investigations of POA which record continuous signals have detected apnea with manual scoring by a single expert. Thus, we felt it would be of interest to compare the performance of AUREA and a single scorer (IS) using reference data obtained from the EM estimate of 5 scorers. Overall accuracy for AUREA was 0.80, whereas for IS it was 0.90. Across the 21 cases, the mean accuracy was higher for IS compared with AUREA (Fig 15). This was not surprising as the EM reference was a consensus of the manually scored records and as shown in Figs 4, 5 and 7 the thresholds for classification differed for the EM and AUREA.

Pattern specific performance for AUREA and IS are given in Table 7. The F-scores for the classification of PAU by AUREA and IS, were respectively 0.56 and 0.70. Precision for PAU was higher for AUREA; thus the likelihood that a PAU classification was correct (positive predictive value) was greater with AUREA. Recall was higher with IS, than AUREA; thus, sensitivity with AUREA was lower compared with IS.

The pattern specific classifications for AUREA and IS were similar. (Fig 16) AUREA classified more PAU and ASB than IS, whereas IS classified more SYB than AUREA. (Table 7)

## Discussion

In this work we describe AUREA, an automated analysis system designed to classify breathing patterns in infants from signals obtained with dual belt RIP. In contrast to other published

**Table 4. Confusion matrix.** Confusion matrix expressed as n samples (percent total pattern) classified by the Automated Unsupervised Respiratory Event Analysis (AUREA) with those classified by Expectation Maximization (EM).

| True Patterns classified by EM | Predicted Patterns classified by AUREA | | | | Pattern Specific Confusion |
|---|---|---|---|---|---|
| | PAU | SYB | ASB | UNKNOWN | |
| PAU | 827,708 (77.8%) | 60,918 (5.7%) | 14,198 (1.3%) | 161,609 (15.2%) | 22.2% |
| SYB | 258,138 (1.9%) | 11,216,048 (83.7%) | 402,054 (3.0%) | 1,527,241 (11.4%) | 16.3% |
| ASB | 69,212 (4.4%) | 42,649 (2.72%) | 1,189,612 (75.8%) | 267,963 (17.07%) | 24.2% |
| UNKNOWN | 533,542 (8.24%) | 646,939 (9.99%) | 473,037 (7.30%) | 4,825,112 (74.50%) | 25.5% |

ASB = asynchrony, PAU = pause, SYB = synchronous.

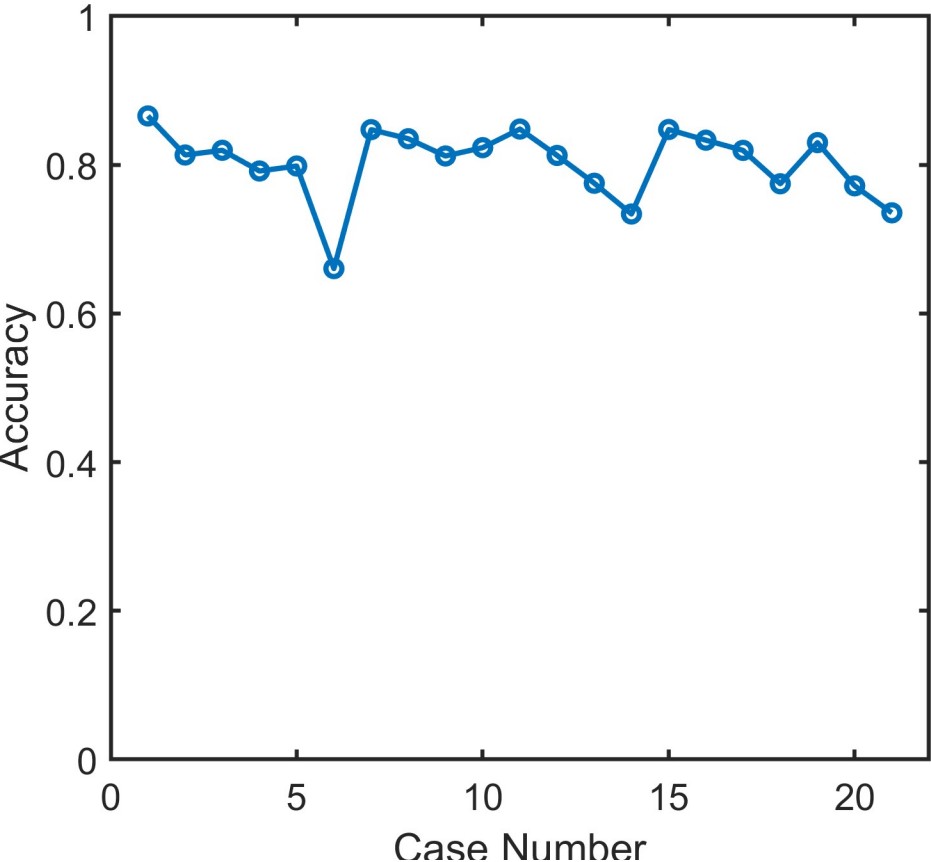

**Fig 11. Accuracy.** Overall accuracy across the 21 records classified by the automated unsupervised respiratory event analysis (AUREA).

automated analysis systems, classification by AUREA was unsupervised and made no assumptions of quasi-sinusoidal waveforms [34, 35]. Our approach differed from that of the AASM as all data samples were classified for all observed infant behaviors. Concatenation of contiguous samples assigned identical patterns then allowed an analysis of pattern events.

The metrics used to classify the PAU pattern were designed to comply with the AASM scoring rule #5.2, for pediatric apnea [14]. The PAU pattern exhibited very low normalized variance, a feature consistent with this AASM scoring rule. The AUREA patterns had properties consistent with the AASM definitions for breathing and apnea in infants. Moreover, the PAU, ASB and SYB patterns assigned by AUREA were in good agreement with the patterns assigned by the human scorers. As discussed in the previous section, the breathing patterns of SYB and ASB had a respiratory frequency and phase that were within the range of published values for infants.

Overall, AUREA classified samples with a mean accuracy of 0.8 when evaluated with respect the reference data derived from expectation maximization. Pattern specific precision, recall and F-score for SYB were all $>0.8$. The F-score for PAU was 0.60; precision was 0.78, and recall was 0.49. Pattern matching for PAU events was excellent and in 97% of the EM classified PAU events$>2$s, 50% of samples were also assigned PAU by AUREA. Moreover, in 92% of the PAU events $>14$s detected by AUREA the majority ($>50$%) of samples were also assigned PAU by EM. This excellent pattern matching lends credibility to the exploratory analysis

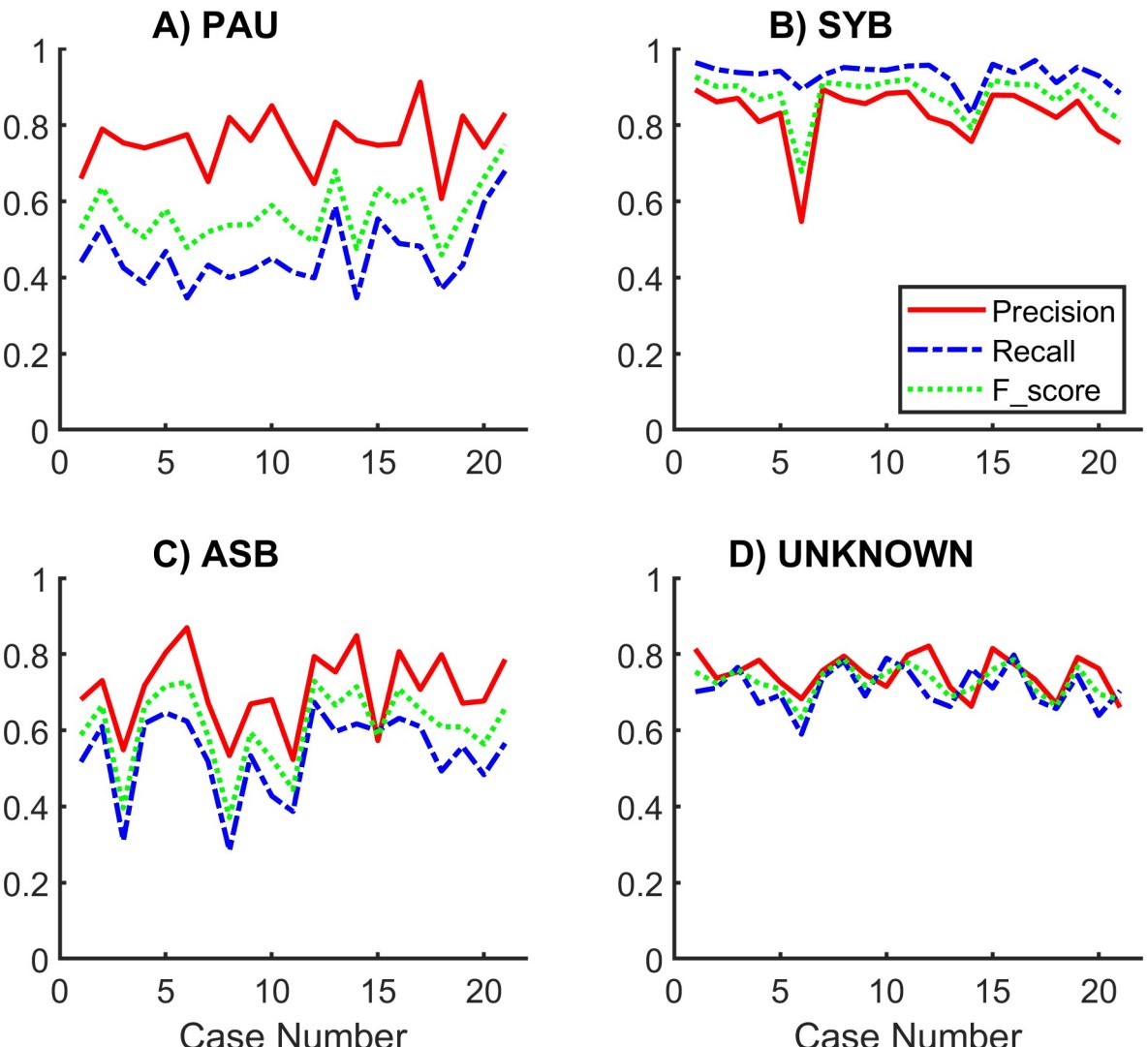

**Fig 12. Pattern specific performance indices.** Pattern specific mean performance indices for precision, recall and F-score across the 21 records classified by the automated unsupervised respiratory event analysis (AUREA).

reported by Robles-Rubio et al. [20]. The PAU specific recall of 0.49 and an F-score of 0.60, suggest that a performance evaluation of PAU pattern events is warranted.

We believe that two main reasons contributed to the differences between the EM and AUREA pattern classification. Firstly, the EM reference was not perfect ground truth data.

**Table 5. Pattern specific performance indices.** Sample by sample mean performance indices for the four breathing patterns classified by AUREA (Automated Unsupervised Respiratory Event Analysis). The UNKNOWN pattern combined the MVT, SIG and UNK patterns.

| Breathing Pattern | Precision | Recall | F score |
|---|---|---|---|
| PAU | 0.78 | 0.49 | 0.60 |
| SYB | 0.84 | 0.94 | 0.88 |
| ASB | 0.76 | 0.57 | 0.65 |
| UNKNOWN | 0.75 | 0.71 | 0.73 |

ASB = asynchrony, MVT = movement, PAU = pause, SIG = sigh, SYB = synchronous, UNK = unknown.

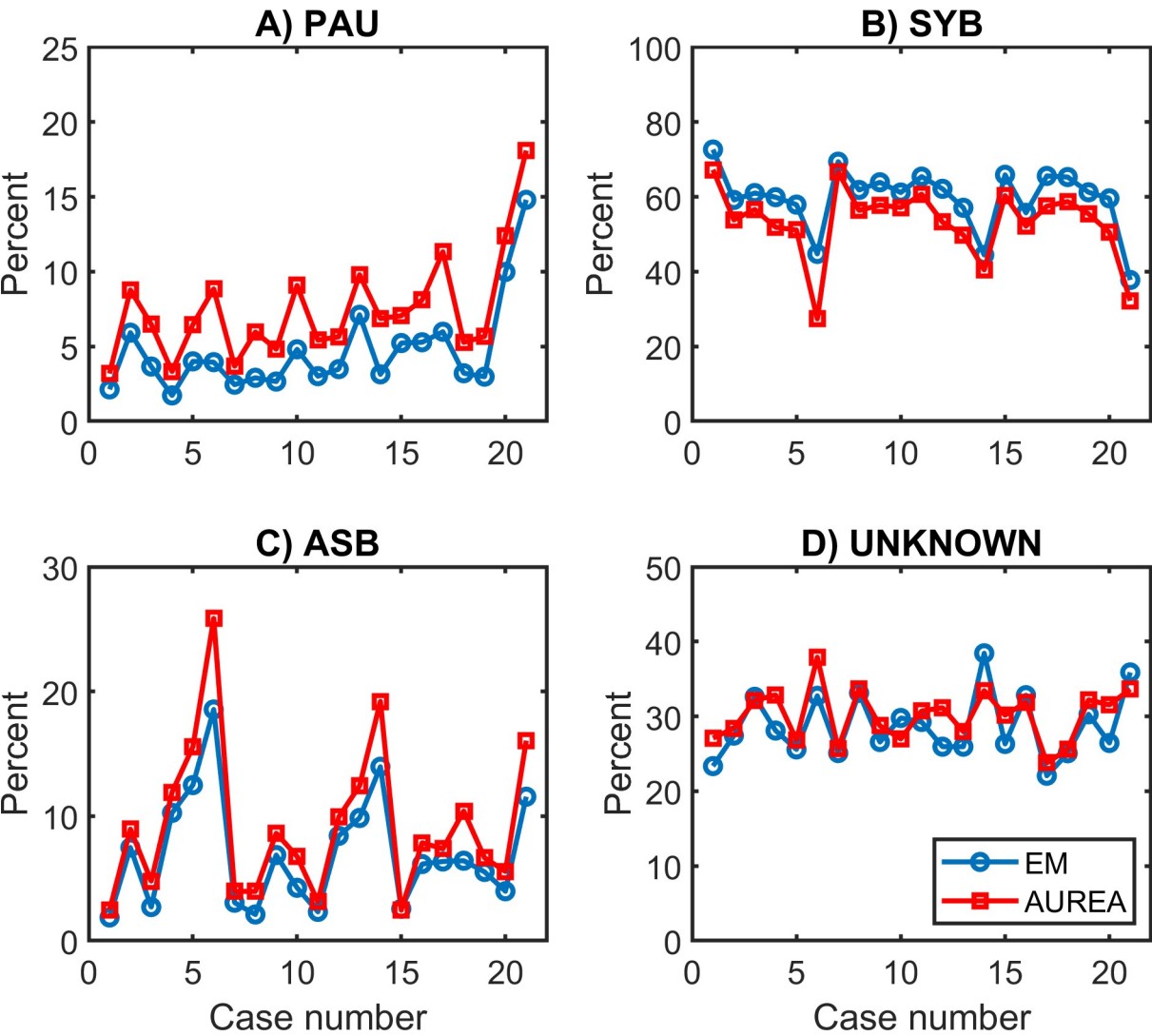

**Fig 13. Distribution of samples across cases.** The percent samples classified to each pattern by both AUREA and EM across the 21 cases. ASB = asynchrony, AUREA = automated unsupervised respiratory event analysis, EM = expectation maximization, PAU = pause, SYB = synchronous.

Although, the EM record, by evaluating and weighting pattern specific scorer performance, in theory, should have provided an optimal solution, the probability that the EM estimate was correct was, in fact, not 100%, as reported in Table 4. Manual scoring is error prone [16] and classification by the human scorer exhibits both inter and intra-scorer variability [21]. Although the AASM regards manual scoring as the gold standard method for analysis of RIP signals, [14] the record scored by trained human experts does not produce perfect ground truth data.

Secondly, the classification procedure differed for AUREA and manual scoring. AUREA was informed exclusively by the metrics derived from the signals of RCG and ABD. In contrast manual scoring was informed by the raw signals of RCG and ABD, the Lissajous loops, and the signals PPG and blood oxygen saturation. Information in these latter two signals may have influenced the patterns assigned during the manual scoring. Finally, whereas AUREA

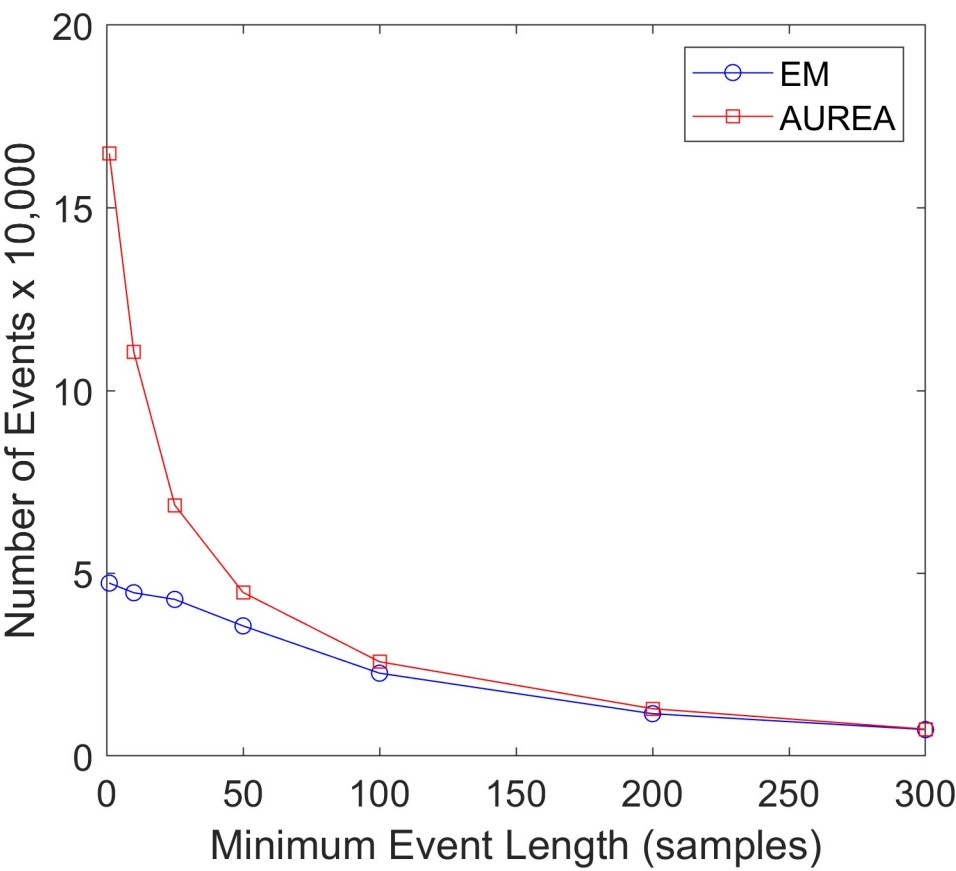

**Fig 14. Fragmentation.** The effect of event length on fragmentation for the Automated Unsupervised Respiratory Event Analysis (AUREA) and the reference record obtained by Expectation Maximization (EM).

evaluated each sample independently, the human expert may have been informed by breathing patterns preceding and following a segment of data.

In the majority of samples classified as PAU, SYB and ASB the probability that the final EM estimate was correct was $\geq$ 95% (Table 4). Thus although not perfect, we consider the EM record to be the best available reference with which to evaluate AUREA. Comparing the AUREA and EM classifications, it was evident in Figs 4 and 5 that the thresholds employed by the binary k-means classifiers provided a better separation of PAU, and MVT, from the remaining samples, and in Fig 7 between SYB, and ASB.

**Table 6. Pattern matching.** Pattern matching between samples classified by Expectation Maximization (EM) and AUREA (Automated Unsupervised Respiratory Event Analysis). At event lengths >2s pattern matching was highest for PAU events.

| Pattern event>2s | Fraction of EM determined pattern events with 50% of samples assigned identical patterns by AUREA |
|---|---|
| PAU events | 0.97 |
| SYB events | 0.78 |
| ASB events | 0.76 |
| UNKNOWN events | 0.74 |

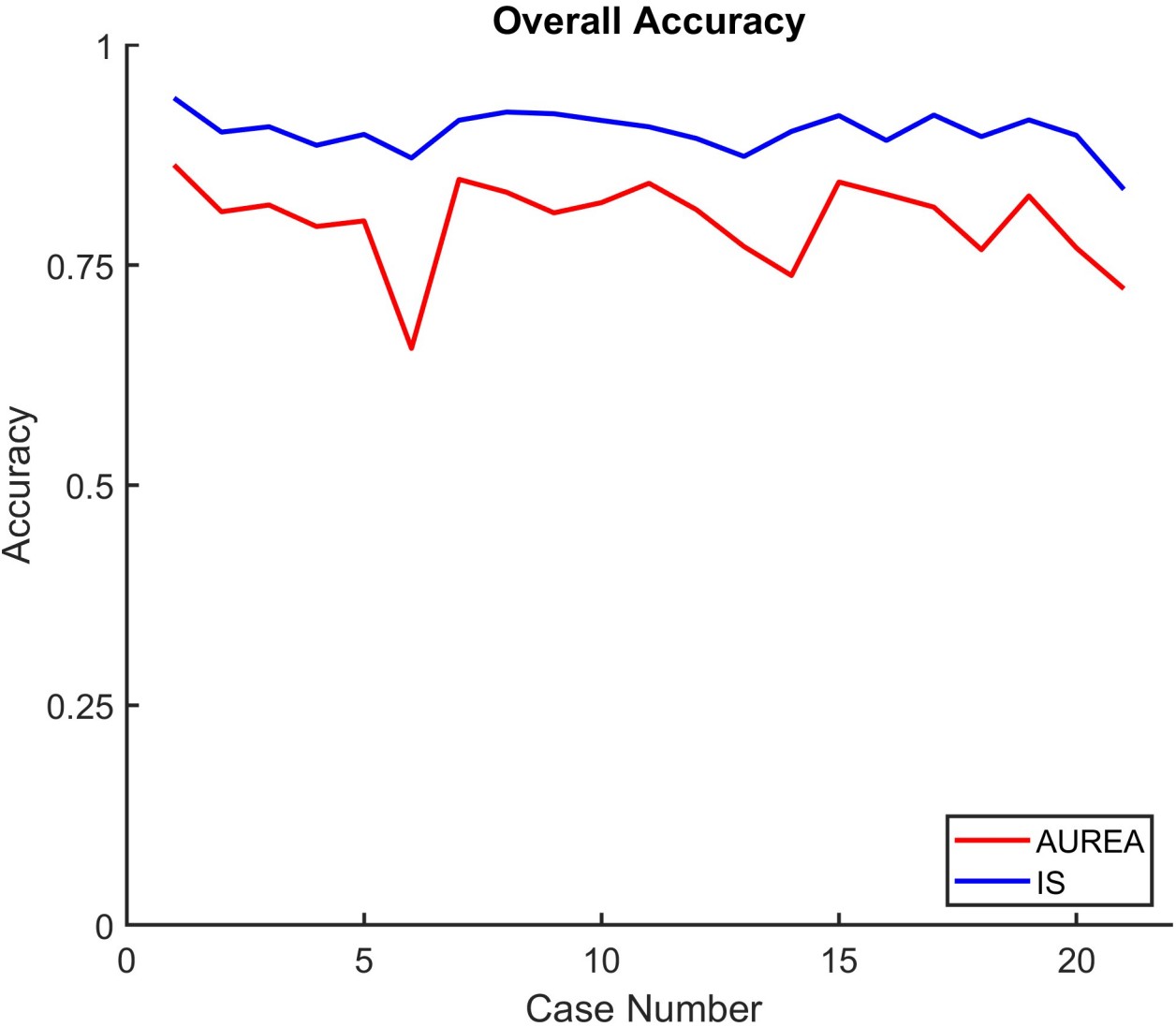

**Fig 15. Accuracy for AUREA and IS.** Accuracy across the 21 records classified by the automated unsupervised respiratory event analysis (AUREA) and individual scorers (IS). (The EM reference data were obtained from the combination of 5 scorers.).

## Limitations

A limitation of the current study was the assumption of a homogeneous population of infants; an assumption that allowed us to train AUREA with the entire dataset. The percentages of breathing patterns across records were then averaged and used to adjust the boundaries for

**Table 7. Pattern specific performance.** Mean pattern specific performance classification by AUREA (Automated Unsupervised Respiratory Event Analysis) versus IS (individual scorer).

|  | Precision | | Recall | | F-score | | Patterns (% Total) | |
|---|---|---|---|---|---|---|---|---|
|  | AUREA | IS | AUREA | IS | AUREA | IS | AUREA | IS |
| PAU | 0.75 | 0.64 | 0.45 | 0.80 | 0.56 | 0.70 | 7.4 | 3.9 |
| SYB | 0.83 | 0.96 | 0.93 | 0.92 | 0.88 | 0.94 | 53.2 | 62.6 |
| ASB | 0.71 | 0.74 | 0.53 | 0.82 | 0.60 | 0.77 | 9.2 | 6.4 |
| UNKNOWN | 0.75 | 0.84 | 0.71 | 0.89 | 0.72 | 0.86 | 30.1 | 27.1 |

ASB = asynchronous, PAU = pause, SYB = synchronous.

Table of symbols.

| Symbol | Definition | Default Value |
|---|---|---|
| $ABD$ | The abdomen RIP signal | |
| $b^+$ | Synchronous breathing metric | |
| $b^-$ | Asynchronous breathing metric | |
| $f_{RESP}$ | Respiratory frequency | |
| $f_s$ | Sampling rate (Hz) | |
| $N_{MA}$ | Window width used to compute $RIP_{MA}$ | |
| $npp_{RIP}$ | Nonperiodic power metric for RIP | |
| $N_{qv}$ | Window width used to estimate quantiles of $V_{RIP}(n)$ | |
| $N_{QRMS}$ | Window width used to estimate $rms^Q_{RIP_{MA}}$ | |
| $N_{RMS}$ | Window width used to estimate rms values | |
| $N_{SMO}$ | Window width used to smooth RIP signal | |
| $N_{LF}$ | Window width used to remove low frequency components of RIP signals | |
| $N_V$ | Window width used to estimate normalized variance metric | |
| $NV_{RIP}$ | Normalized variance metric of RIP | |
| $\Phi$ | Phase between RCG and ABD | |
| $q$ | Quantile used to normalize $V_{RIP}$ | |
| $RCG$ | The ribcage RIP signal | |
| $RIP$ | Respiratory inductance plethysmography signal | |
| $RIP_B$ | Binary version of RIP signal | |
| $RIP_{MA}$ | Moving average value of RIP signal | |
| $RIP_{RAW}$ | Raw RIP signal | |
| $RIP_{SMO}$ | Smoothed RIP signal | |
| $RIP_{LF}$ | Preprocessed RIP signal with low frequency components removed | |
| $rms_{RIP_{MA}}$ | Root mean square value of $RIP_{MA}$ | |
| $rms^Q_{RIP_{MA}}$ | $q^{th}$ quantile of $rms_{RIP_{MA}}$ | |
| $V_{RIP}$ | Instantaneous variance estimate of the RIP signal | |
| $V^Q_{RIP}$ | is the $q^{th}$ quantile of the most recent $N_{QV}$ samples ($N_{QV}>>N_V$) | |

unbalanced proportions to similar extent for all records. The results show that the distribution of breathing patterns differed across the 21 records (Fig 13). These differences may have influenced the adjustment of boundaries and thereby, the classification performance of AUREA. Indeed, Fig 11 shows that accuracy was lowest for cases 6, 14, and 21, the very same cases with the greatest differences in pattern distribution. Future work might consider alternate approaches to mitigate the problem of sample unbalance.

A second limitation was that we did not develop a metric to classify sighs; comprising 1.7% of all samples (Table 2). The confusion matrix in Table 3 shows that AUREA patterns were confused most often with UNKNOWN, a pattern that included sighs. Future work should develop classifier to detect sighs as it may improve the classification performance of AUREA. In addition, infants' sighs are known to co-localize with apnea[5, 36]. A systematic study of the relationship between sighs and apnea in infants at risk for POA might be informative.

AUREA has several advantages; high accuracy (0.8), and a precision above 0.72 for all four breathing patterns; PAU (0.78), SYB (0.84), ASB (0.76) and UNKNOWN (0.74). The excellent PAU specific pattern matching suggests that AUREA will prove useful for the detection of clinically relevant respiratory pauses, including apnea. In the study of POA, manual scoring by a single expert has been the usual method for the analysis of breathing patterns [5, 6, 11, 13].

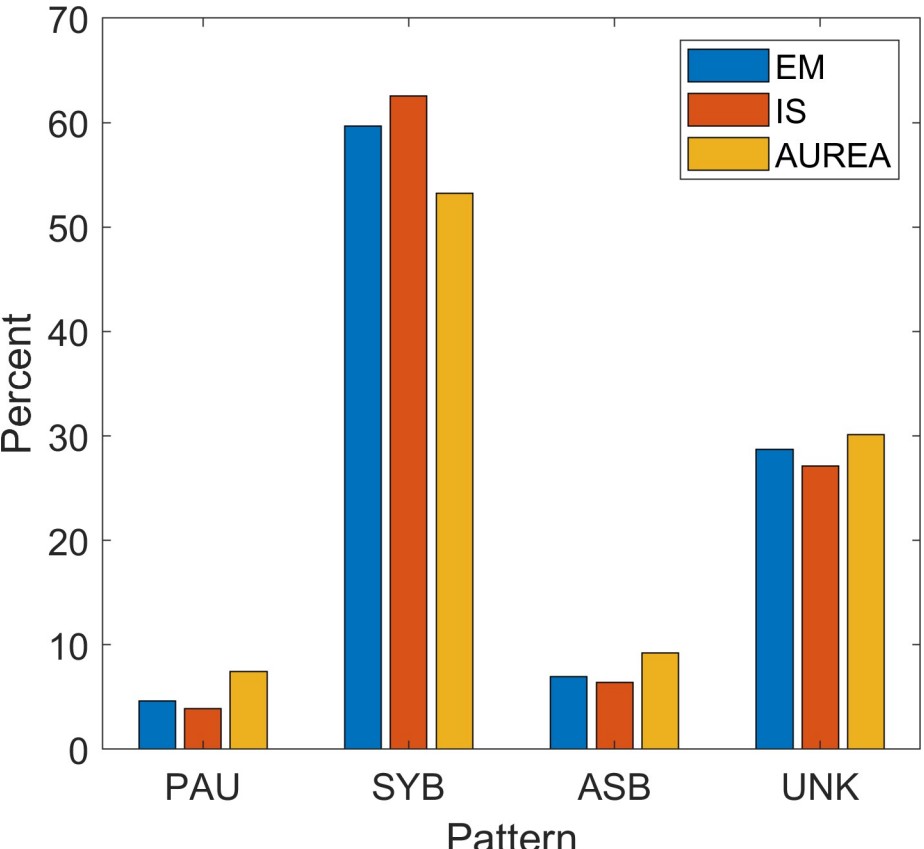

**Fig 16. Classification of patterns by AUREA, IS and EM.** Classification of pattern in samples obtained from expectation maximization (EM), AUREA (Automated Unsupervised Respiratory Event Analysis) and individual human scorers (IS). Samples classified MVT, UNK and SIH were combined into a single group UNKNOWN. ASB = asynchrony, MVT = movement, PAU = pause, SIH = sigh, SYB = synchronous, UNK = unknown.

However manual analysis exhibited intra-and inter-scorer variability and was time consuming; requiring several weeks to analyze the 21 files. In contrast AUREA had perfect repeatability and the speed of analysis was very fast; requiring less than a minute to analyze the 21 files.

## Conclusion

The demographic risk factors for POA, namely gestational age, age at the time of surgery, and a history of apnea, have been known for decades [3, 8]. Although POA is attributed to the effect of anesthesia on an immature cardiorespiratory system, a biomarker of this immaturity has yet to be identified. An impediment to the systematic study of POA has been the lack of an accurate and timely method to classify the breathing patterns recorded in cardiorespiratory signals. The evidence presented in the current work suggests that AUREA can be used as the method to rapidly classify breathing patterns, sample by sample, in signals recorded from dual belt RIP. For example, Robles-Rubio et al. evaluating pause events classified by AUREA reported risk factors, including anesthetic agents, associated with infants who exhibited respiratory pauses in excess of 14.6s [20]. Furthermore, AUREA pattern classifications have also been used to evaluate the breathing patterns of premature infants following extubation of the trachea [37, 38]. Both AUREA patterns and AUREA metrics will be used in the development of a predictor of extubation readiness in the extreme premature infant [37]. Thus the analyses

of patterns and metrics derived from AUREA might reveal breathing behaviours that identify the vulnerable infant at increase risk for POA.

## Author Contributions

**Conceptualization:** Carlos A. Robles-Rubio, Robert E. Kearney, Karen A. Brown.

**Data curation:** Carlos A. Robles-Rubio, Robert E. Kearney, Gianluca Bertolizio, Karen A. Brown.

**Formal analysis:** Robert E. Kearney, Karen A. Brown.

**Investigation:** Carlos A. Robles-Rubio.

**Methodology:** Carlos A. Robles-Rubio, Robert E. Kearney, Gianluca Bertolizio, Karen A. Brown.

**Project administration:** Robert E. Kearney.

**Resources:** Robert E. Kearney.

**Software:** Carlos A. Robles-Rubio, Robert E. Kearney.

**Supervision:** Robert E. Kearney, Karen A. Brown.

**Validation:** Robert E. Kearney, Karen A. Brown.

**Visualization:** Robert E. Kearney, Karen A. Brown.

**Writing – original draft:** Robert E. Kearney, Karen A. Brown.

**Writing – review & editing:** Carlos A. Robles-Rubio, Robert E. Kearney, Gianluca Bertolizio, Karen A. Brown.

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
