## [Decision Letter · Decision Letter 0]

8 Jul 2020

PONE-D-20-09601

Automatic Unsupervised Respiratory Analysis of Infant Respiratory Inductance Plethysmography Signals

PLOS ONE

Dear Dr. Karen Ann Brown,

Thank you for submitting your manuscript to PLOS ONE. After careful consideration, we feel that it has merit but does not fully meet PLOS ONE’s publication criteria as it currently stands. Therefore, we invite you to submit a revised version of the manuscript that addresses the points raised during the review process.

We look forward to receiving your revised manuscript.

Kind regards,

Wajid Mumtaz

Academic Editor

PLOS ONE

Journal Requirements:

Reviewers' comments:

Reviewer's Responses to Questions

**Comments to the Author**

1. Is the manuscript technically sound, and do the data support the conclusions?

Reviewer #1: Yes

2. Has the statistical analysis been performed appropriately and rigorously? 

Reviewer #1: N/A

3. Have the authors made all data underlying the findings in their manuscript fully available?

Reviewer #1: Yes

4. Is the manuscript presented in an intelligible fashion and written in standard English?

Reviewer #1: Yes

5. Review Comments to the Author

Reviewer #1: The study proposes an automatic system for classifying breathing patterns in infants. The proposed system is called Automated Unsupervised Respiratory Event Analysis (AUREA) and its goal is to recognize five respiratory events: pause, movement, synchronous breathing, asynchronous breathing and unknown.

In the front-end stage, the system uses ribcage and abdomen signals from dual-belt respiratory inductance plethysmography to extract a set of features representing a log-normalized variance, non-periodic power, synchronous/asynchronous breathing metrics and respiratory frequency and phase. The backend is implemented by combining a set of binary k-means classifiers into a decision tree. In the top node, the incoming data segment is classified into pause vs. non-pause classes. Segments classified as non-pause progress to a lower node where they are classified as movement vs. non-movement. If non-movement, they are then classified into synchronous breathing vs. non-synchronous classes. Non-synchronous are then classified into asynchronous vs. unknown.

AUREA works in a two-stage fashion, where the same sequence of data is first sequentially clustered, in each step to two clusters representing the target class and the anti-class. Input data transcriptions/labels are not provided to the system. Instead, the discovered clusters are assigned class labels based on assumptions on ‘how the data should look’ for each class – for example that a specific feature should exhibit low variance in pause segments and high variance for non-pause segments. Once the whole training set is utilized to build the decision tree classifier, the training set is passed again to the tree, this time for classification.

Data from 21 (16 male, 5 female) infants, where 19 of them were formerly premature infants and thus more prone to postoperative apnea, for evaluations of the proposed scheme, totaling in 116 hours of data. As a reference, manual transcriptions from 3 transcribers are used, where each transcriber had to transcribe each recording twice. Manual transcriptions were combined into a single reference track using an EM-PSEQ (Expectation-Maximization Pattern of Sequence) method established by the authors in their prior work.

The topic of the study, design of an automated system that may help detecting apnea in infants, is current and of high interest to the community. The authors built upon their previous work on this topic and the outcomes may be quite interesting to the reader. This being said, there are several issues the authors should address prior to publication.

Some notation and terminology in the paper is ambiguous or seems to be used in a confusing manner:

- The authors repeatedly use the term ‘gold standard’ while referring to the reference data used in evaluations of their system. This term is routinely used with reference *methods* (i.e., methods that are known to perform well and can be used to establish a performance standard). On the other hand, ground truth is used for reference data (i.e., the data a system output will be compared to). The authors should replace all references to gold standard data by ground truth data. Moreover, there is no need to use quotation marks when referring to gold standard methods or ground truth data.

- The authors consistently use the term two-sided window to emphasize that the window spans across both the preceding and succeeding samples around the current sample. However, this term sounds somewhat unusual and confusing, since the window is a standard rectangular window and every window has two ‘sides’ (the left and the right one). Analysis windows are defined by type/shape and start/end points. Sometimes a window is defined by a starting point and length. In other times, the window is defined by its center point and length – which seems to be the case of this study. It is sufficient for the authors to mention (once) that the window is centered around the discrete time n (which is also clearly stated in all the window processing formulas) and later on just refer to it as an analysis window rather that two-sided window (as the reader would have to constantly remind themselves that it is indeed a regular rectangular window, which drives attention away from the actual focus of the manuscript).

- How was the structure of the decision tree determined (i.e., the order of the class/anti-class pairs)? Did you try reordering the sequence of the binary classifiers?

- Page 8-9: “PAU has the highest priority, forcing the other pattern detectors to zero.” This is a bit hard to read – forcing a detector to zero is not a usual way do say that a detector is disengaged. It would be helpful to describe your set of detectors as a decision tree and explain that the detectors are engaged sequentially as the observation descends through the tree. If the decision is set to the target class, the tree reached a leaf and the classification is over. If the decision is set to the anti-class, the sample descends to the lower tree node where another binary classification is performed, etc.

- Preprocessing of signals - Page 9: “The RIP signals were pre-processed to remove low frequency trends” – using the term low frequency ‘components’ would be more standard and easier to read; moreover, the described method performs segment-level mean normalization (subtracting the segmental mean), so it would be more accurate to say ‘to remove the DC and low frequency components’

- Page 9 – “detrended RIP signals” – signals with removed DC and low frequency components, or simply lowpass filtered signals; there are certainly some trends left in the signals, otherwise they would be useless – so ‘detrended’ is an exaggeration (it may also give a negative vibe)

- Page 10 – “estimating the sample-by-sample variance” – this is rather puzzling as it may read as if variance was calculated for each sample. Variance can be only calculated for a segment of samples, as, unlike mean, it is undefined for a single sample. It appears from the text that the analysis window step (skip rate) was one sample. So there would be an updated variance estimate for every new window and we are getting a new window with a single sample step, but that doesn’t mean that we are measuring variance of single samples – please rephrase to avoid confusion.

- Page 11 – “the RIP signal will contain stochastic periodic noise” – the terms ‘stochastic’ and ‘periodic’ contradict each other. A periodic signal will not be random if it is periodic – seeing one period of the signal means we know everything about it, while samples of a stochastic signal are random – please clarify/rephrase.

- What is the analysis window length in your system? Please provide this information in the text with elaboration on how the length was chosen to accommodate the modeling/classification process.

- Page 7 – “These reference segments, totaled 2000s per second” – what does 2000s refer to? I would assume samples (i.e., sampling rate of 2kHz), but your sequence reads as if there were ‘2000s’ reference segments per second which is indeed not the case; please clarify/rephrase.

- References to figures – please use consistent format for figure references – ideally Fig. <number>. Currently, the figure references randomly switch between upper/lower-cases (Fig 16, fig 4), spaces or no spaces (Fig13).

- References to literature at the end of the sentence would be typically included as a part of the sentence (i.e., before the closing period) – such as “and tend to zero otherwise [27].” There is always a space character between the opening square bracket and the previous word. In your text, the end of sentence literature references are mostly placed behind the closing period (e.g., “…at the respiratory frequency.[27]”) which is highly unusual and makes the following sentence basically start with the reference (this style is sometimes used when the literature references are typeset as upper indices). Also, elsewhere your text does include the references before the period (e.g., “most of PAU and MVT power is located between 0 Hz and 0.4 Hz [19, 24].” or “tend to zero otherwise [27].”) Please unify the style.

- The language is clear and easy to follow, but there are numerous typos in the text – mostly missing spaces between adjacent words or words and references or ends of sentences and beginnings of the following ones.

- …et al. – the period behind al. is omitted in multiple places

- Figures with multiple trends – Fig 4-7, 12, 15 – please use different types of trend lines to distinguish the trends not just by color – if printed b/w or for readers with color vision deficiency, the current format might be confusing.</number>

6. PLOS authors have the option to publish the peer review history of their article (what does this mean?). If published, this will include your full peer review and any attached files.

Reviewer #1: No

---

## [Decision Letter · Decision Letter 1]

17 Aug 2020

Automatic Unsupervised Respiratory Analysis of Infant Respiratory Inductance Plethysmography Signals

PONE-D-20-09601R1

Dear Dr. Karen Ann Brown,

We’re pleased to inform you that your manuscript has been judged scientifically suitable for publication and will be formally accepted for publication once it meets all outstanding technical requirements.

Kind regards,

Wajid Mumtaz

Academic Editor

PLOS ONE

Additional Editor Comments (optional):

Reviewers' comments:

Reviewer's Responses to Questions

**Comments to the Author**

1. If the authors have adequately addressed your comments raised in a previous round of review and you feel that this manuscript is now acceptable for publication, you may indicate that here to bypass the “Comments to the Author” section, enter your conflict of interest statement in the “Confidential to Editor” section, and submit your "Accept" recommendation.

Reviewer #1: All comments have been addressed

2. Is the manuscript technically sound, and do the data support the conclusions?

Reviewer #1: Yes

3. Has the statistical analysis been performed appropriately and rigorously? 

Reviewer #1: Yes

4. Have the authors made all data underlying the findings in their manuscript fully available?

Reviewer #1: Yes

5. Is the manuscript presented in an intelligible fashion and written in standard English?

Reviewer #1: Yes

6. Review Comments to the Author

Reviewer #1: The authors have addressed all my comments in the reviewed manuscript. I believe the manuscript is now in a good shape to be published as is.

7. PLOS authors have the option to publish the peer review history of their article (what does this mean?). If published, this will include your full peer review and any attached files.

Reviewer #1: No

---

## [Editor Report · Acceptance letter]

2 Sep 2020

PONE-D-20-09601R1 

Automatic Unsupervised Respiratory Analysis of Infant Respiratory Inductance Plethysmography Signals 

Dear Dr. Brown:

I'm pleased to inform you that your manuscript has been deemed suitable for publication in PLOS ONE. Congratulations! Your manuscript is now with our production department. 

Kind regards, 

on behalf of

Dr. Wajid Mumtaz 

Academic Editor

PLOS ONE